# Structural basis for the H2AK119ub1-specific DNMT3A-nucleosome interaction

Xinyi Chen[1,7], Yiran Guo[2,3,7], Ting Zhao[4], Jiuwei Lu [1], Jian Fang [1], Yinsheng Wang[4,5], Gang Greg Wang [2,3,6] ✉ & Jikui Song [1] ✉

Isoform 1 of DNA methyltransferase DNMT3A (DNMT3A1) specifically recognizes nucleosome monoubiquitylated at histone H2A lysine-119 (H2AK119ub1) for establishment of DNA methylation. Mis-regulation of this process may cause aberrant DNA methylation and pathogenesis. However, the molecular basis underlying DNMT3A1–nucleosome interaction remains elusive. Here we report the cryo-EM structure of DNMT3A1's ubiquitin-dependent recruitment (UDR) fragment complexed with H2AK119ub1-modified nucleosome. DNMT3A1 UDR occupies an extensive nucleosome surface, involving the H2A-H2B acidic patch, a surface groove formed by H2A and H3, nucleosomal DNA, and H2AK119ub1. The DNMT3A1 UDR's interaction with H2AK119ub1 affects the functionality of DNMT3A1 in cells in a context-dependent manner. Our structural and biochemical analysis also reveals competition between DNMT3A1 and JARID2, a cofactor of polycomb repression complex 2 (PRC2), for nucleosome binding, suggesting the interplay between different epigenetic pathways. Together, this study reports a molecular basis for H2AK119ub1-dependent DNMT3A1–nucleosome association, with important implications in DNMT3A1-mediated DNA methylation in development.

DNA methylation provides a critical epigenetic mechanism for ensuring appropriate gene silencing, imprinting and genome stability[1–4]. In mammals, establishment of DNA methylation is mainly achieved by de novo DNA methyltransferases DNMT3A and DNMT3B[5]. Precise regulation of DNMT3A/DNMT3B functions across the genome is essential for orchestrating a proper DNA methylation landscape, which governs cell fate determination and development. Mutation of DNMT3A is linked to cancers (e.g. acute myeloid leukemia and paraganglioma) and developmental disorders (e.g. Tatton-Brown-Rahman syndrome and microcephalic dwarfism)[6–10]. However, the mechanisms underlying the DNMT3A-mediated epigenetic regulation remain elusive, the improved understanding of which shall help to develop novel therapeutic strategies against those human diseases caused by the DNMT3A malfunction.

Human *DNMT3A* gene encodes at least two different protein isoforms, namely DNMT3A1 and DNMT3A2, with the latter lacking a 223-amino-acid-long N-terminal region that the full-length DNMT3A1 has[11]. It has been established that DNMT3A1 and DNMT3A2 have distinctive expression patterns during development. DNMT3A2 expression dominates in germ cells and embryonic stem cells (ESCs), but gradually decreases as development progresses, except in tissues like spleen, thymus and testis[12], and cancer cells[13]. In contrast, DNMT3A1 is widely and stably expressed in somatic cells[14]. Also different is their subcellular localization patterns: DNMT3A1 was found to be exclusively localized within the nuclei and enriched in transcriptionally inactive heterochromatin, whereas DNMT3A2 exhibited a diffused pattern in the nuclei, strongly associated with transcriptionally active euchromatin[11,14].

[1]Department of Biochemistry, University of California, Riverside, CA 92521, USA. [2]Department of Pharmacology and Cancer Biology, Duke University School of Medicine, Durham, NC 27710, USA. [3]Duke Cancer Institute, Duke University School of Medicine, Durham, NC 27710, USA. [4]Environmental Toxicology Graduate Program, University of California, Riverside, CA 92521, USA. [5]Department of Chemistry, University of California, Riverside, CA 92521, USA. [6]Department of Pathology, Duke University School of Medicine, Durham, NC 27710, USA. [7]These authors contributed equally: Xinyi Chen, Yiran Guo. ✉e-mail: greg.wang@duke.edu; jikui.song@ucr.edu

The molecular basis underlying the distinct cellular localization and functionalities of DNMT3A1 and DNMT3A2 remains unclear.

DNMT3A-directed DNA methylation is tightly controlled by a coordinated action between the C-terminal methyltransferase (MTase) domain and the N-terminal regulatory region[15–17]. It has been reported that the MTase domain can mediate CpG-specific methylation in either a homo-tetrameric or hetero-tetrameric form[13,18–24]. Meanwhile, the N-terminal Proline-Tryptophan-Tryptophan-Proline (PWWP) domain recognizes histone H3 lysine 36 dimethylation (H3K36me2) and tri-methylation (H3K36me3)[25–28], which directs the recruitment of DNMT3A to those H3K36me2/3-enriched heterochromatic and inter-genic regions[27,29]. Furthermore, the ATRX-DNMT3A-DNMT3L (ADD) domain within the N-terminal regulatory region reads the unmodified histone H3 lysine 4 (H3K4me0)[30,31], which allosterically activates DNMT3A to ensure H3K4me0-specific DNA methylation[20,32]. In addi-tion, recent studies have identified that the N-terminal domain unique to DNMT3A1 harbors an ubiquitin-dependent recruitment (UDR) region that interacts with nucleosomes carrying histone H2A lysine 119 mono-ubiquitylation (H2AK119ub1)[12,28], a repressive histone modifica-tion deposited by polycomb repression complex 1 (PRC1)[33,34]. Such a link between DNA methylation and polycomb signaling is further supported by the observation that numerous gene promoters regu-lated by polycomb gain DNA methylation during neuronal differentiation[35,36]. On the other hand, deletion of the DNMT3A1 N-terminal domain reportedly caused a developmental defect that phenocopies *Dnmt3a1*[−/−] in mice[12]. In paraganglioma[8] and micro-cephalic dwarfism[6] where disease-related mutations specifically target the DNMT3A's PWWP domain, DNA hypermethylation was observed at regions enriched for CpG islands (CGIs), the known targets of poly-comb and H2AK119ub1, suggesting the functional crosstalk between DNMT3A1's PWWP and UDR, as well as the importance of the regula-tion of DNMT3A-nucleosome interaction.

To understand the molecular basis for the interaction between DNMT3A1 and H2AK119ub1-modified nucleosome, we here determine the cryo-EM structure of a complex between the DNMT3A1 UDR and the H2AK119ub1-modified nucleosome core particle (NCP). The struc-ture of DNMT3A1 UDR−H2AK119ub1 NCP complex reveals a multi-faceted interaction: the N-terminal segment of DNMT3A1 UDR forms a U-turn to interact with the H2A-H2B acidic patch of the nucleosome, the central segment occupies the groove formed by H2A C-terminal segment and H3 homodimeric interface, while the C-terminal α-helix contacts both nucleosomal DNA and the ubiquitin moiety. Mutation of the interacting residues of DNMT3A1 leads to reduced or abolished interaction between DNMT3A1 UDR and H2AK119ub1-modified NCP, and compromises the DNA methylation activity of DNMT3A1 on nucleosome substrates in vitro. In mouse ESCs, one tested UDR −nucleosome interaction mutation also leads to strongly impaired chromatin binding and DNA methylation by DNMT3A1. Our combined structural and biochemical analysis further demonstrates that the DNMT3A1 UDR-binding surface of the nucleosome overlaps greatly with the previously observed JARID2-binding site, providing a mechanism for the mutually exclusive interaction with H2AK119ub1 NCP between DNMT3A1 and JARID2, a cofactor of form 2 of polycomb repression complex 2 (PRC2.2)[37,38], and an explanation to the functional antagonism between DNA methylation and PRC2.2-mediated H3 lysine 27 trimethylation (H3K27me3). Together, our study uncovers the molecular basis for the interaction between DNMT3A1 and H2AK119ub1-demarcated chromatin, providing insights into the intri-cate interplays among H2AK119ub1, DNA methylation, and H3K27me3.

## Results
### Structural overview of DNMT3A1 UDR−H2AK119ub1 nucleosome
To elucidate the structural basis for the interaction between the DNMT3A1 UDR and H2AK119ub1-modified nucleosome, we generated a

fragment of DNMT3A1 spanning residues 126–223 (Supplementary Fig. 1a), based on previous reports that this region is responsible for the interaction of DNMT3A1 with H2AK119ub1-modified nucleosome[12,28]. In addition, NCPs were prepared with either wild-type (WT) or K119C-mutated histone H2A, with the latter further conjugated to the G76C-mutated ubiquitin via a dichloroacetone (DCA) linkage[39] (hereafter referred to as H2AK119ub1). Using Biolayer Interferometry (BLI) assay, we observed that the DNMT3A1 UDR binds to the H2AK119ub1-modified NCP with a dissociation constant ($K_d$) of 16.6 nM; in contrast, a -12-fold reduction in binding affinity ($K_d$ of 197 nM) was observed between the DNMT3A1 UDR and the unmodified NCP (Fig. 1a, b). Consistently, our electrophoretic mobility shift assay (EMSA) indicated that an increasing amount of the DNMT3A1 UDR shifted the H2AK119ub1-modified NCP more pronouncedly than the unmodified NCP (Supple-mentary Fig. 1b). Together, these data confirmed previous observations that DNMT3A1's UDR interacts with nucleosome in a H2AK119ub1-specific manner[12,28]. Next, we assembled the complex containing the DNMT3A1 UDR and H2AK119ub1-modified NCP and subsequently determined the cryo-EM structure at 3.26 Å overall resolution (Fig. 1c, d and Supplementary Figs. 2, 3 and Supplementary Table 1).

The structure of the DNMT3A1 UDR−H2AK119ub1 NCP complex reveals that two DNMT3A1 UDR molecules bind to one H2AK119ub1 NCP, with each occupying one histone surface (Fig. 1c, d). We were able to trace the DNMT3A1 residues L170-W210 on one surface but only L170-A192 on the other (Fig. 1c). Coincidently, cryo-EM density for H2AK119ub1-conjugated ubiquitin was only visible on the surface with DNMT3A1 UDR density more clearly defined (Fig. 1e), indicative of conformational flexibility of the corresponding region of DNMT3A1 UDR−H2AK119ub1 NCP complex. Based on these observations, we chose to focus on the histone surface with the better-defined DNMT3A1 UDR fragment for structural analysis. Residues L170-W210 of DNMT3A1 UDR traverse the histone surface, with its N-terminal segment forming a U-turn anchored to the H2A-H2B acidic patch, followed by an extended segment running through the surface groove formed by H2A's C-terminal segment (α3-αC and C-terminal tail) and H3's homodimeric interface, and a C-terminal α-helix sandwiched between nucleosomal DNA and H2AK119C-conjugated ubiquitin (Fig. 1e and Supplementary Fig. 4).

It is worth noting that our sequence analysis of the DNMT3A1 UDR reveals that the H2AK119ub1-interacting segment of DNMT3A1 was highly conserved across evolution (Supplementary Fig. 5), supporting the functional importance of the DNMT3A1 UDR−H2AK119ub1 NCP interaction.

### Structural basis for the interaction between DNMT3A1 UDR and H2A-H2B acidic patch
Detailed structural analysis of the DNMT3A1 UDR−H2AK119ub1 NCP complex further reveals that the U-turn formed by DNMT3A1 R171-R181 sits right on top of the H2A-H2B acidic patch (Fig. 2a−c and Supple-mentary Fig. 4a). Strikingly, the side chain of DNMT3A1 R181 inserts deep into the surface cavity formed by H2A E61, D90, E92 and L93, with the guanidium group of DNMT3A1 R181 engaging in salt-bridge inter-actions with the sidechain carboxylates of H2A E61, D90 and E92 (Fig. 2b and Supplementary Fig. 4b), reminiscent of the acidic patch-anchoring arginine observed for many other nucleosome-binding proteins[40]. Immediately next to the DNMT3A1 R181-binding cavity, the guanidium group of DNMT3A1 R171 is anchored to a shallow cleft formed by H2A E61, E64 and L65 via salt-bridge interactions with the sidechain carboxylates of H2A E61 and E64, as well as van der Waals contacts with the side chain of H2A L93 (Fig. 2c and Supplementary Fig. 4b). At the bottom of the U-turn, DNMT3A1 G175 engages in a hydrogen-bonding interaction with the side chain of H2B Q44 (Fig. 2d). In addition, the indole ring of DNMT3A1 W176 is embedded in the cavity formed by H2A E56 and A60 and H2B V41, Q44, V45 and E110 (Fig. 2d and Supplementary Fig. 4c), the side chain of DNMT3A L180 is

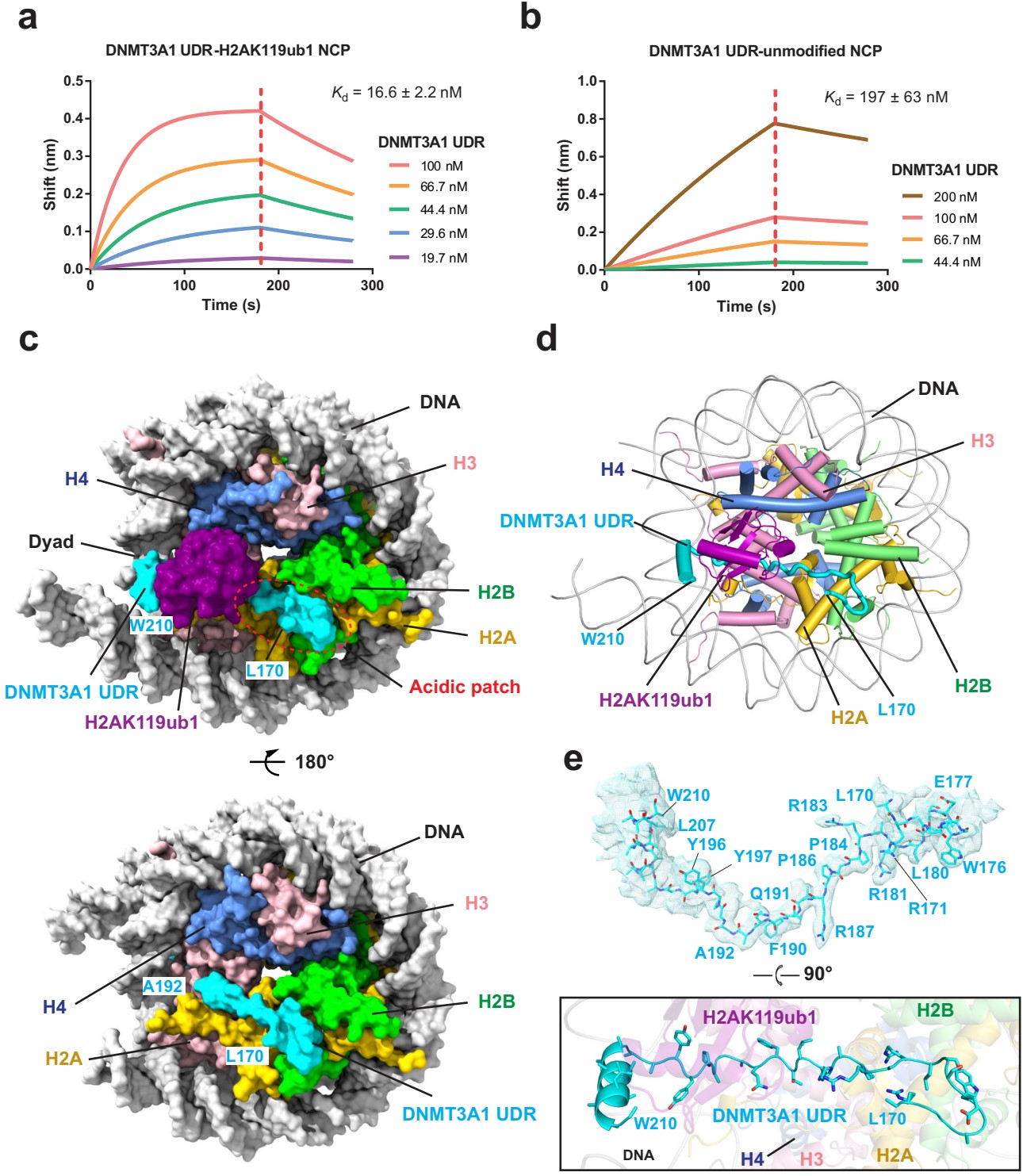

**Fig. 1 | Cryo-EM study of the complex between DNMT3A1 UDR and H2AK119ub1-modified NCP. a**, **b** BLI binding assay for DNMT3A1 UDR with H2AK119ub1-modified (**a**) or unmodified NCP (**b**). Biotinylated NCPs were immobilized on streptavidin XT biosensors, titrated with DNMT3A1 UDR. The various concentrations of DNMT3A1 UDR used for the assay are indicated. The vertical red dashes indicate the time boundaries between the processes of association and dissociation. Data are mean ± s.d. (*n* = 2 biological replicates). One representative set of BLI binding data is shown. **c** Density map for the complex between DNMT3A1 UDR and H2AK119ub1-modified NCP in opposite views, with DNMT3A1 UDR, ubiquitin, nucleosomal DNA,

H2A, H2B, H3 and H4 colored in cyan, purple, gray, gold, lime, pink and corn flower blue, respectively. The region for the H2A-H2B acidic patch is dash circled. Note that DNMT3A1 UDR is not equally traced on the two sides of the histone surface. **d** Atomic model of the complex between DNMT3A1 UDR and H2AK119ub1-modified NCP in cartoon representation. The color scheme is the same as that in (**c**). **e** (Top) Density map for DNMT3A1 UDR with the corresponding atomic model. The N- and C-terminal residues and those with resolved sidechain density are labeled. (Bottom) Atomic model of DNMT3A1 UDR running across the histone surface.

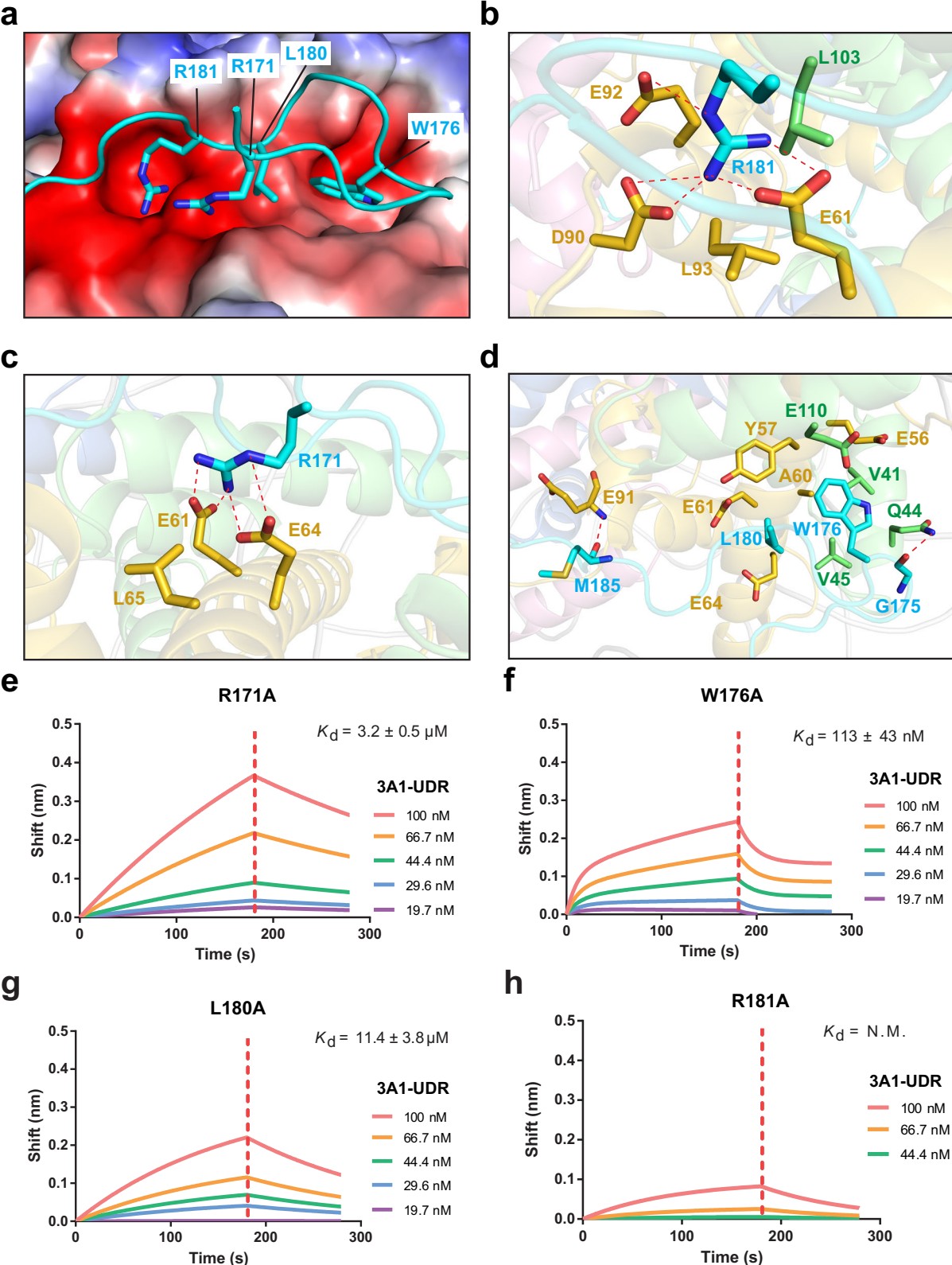

**Fig. 2 | Structural basis for the interaction between DNMT3A1 UDR and H2A-H2B acidic patch. a** Electrostatic surface of the H2A-H2B acidic patch harboring the U-turn segment of DNMT3A1 UDR (cyan). **b, c** Close-up view of DNMT3A R181 (**b**) and R171 (**c**) (cyan) interacting with the H2A (gold)-H2B (lime) acidic patch. Hydrogen-bonding interactions are shown as dashed lines. (**d**) Close-up view of the van der Waals and hydrogen-bonding interactions between additional DNMT3A U-turn residues (cyan) and the H2A (gold)-H2B (lime) acidic patch. **e–h** BLI assays for the interaction between R171A- (**e**), W176A- (**f**), L180A- (**g**) or R181A-mutated (**h**) DNMT3A1 UDR and H2AK119ub1-modified NCP. Data are mean ± s.d. (n = 2 biological replicates). N.M. not measurable. One representative set of BLI binding data is shown.

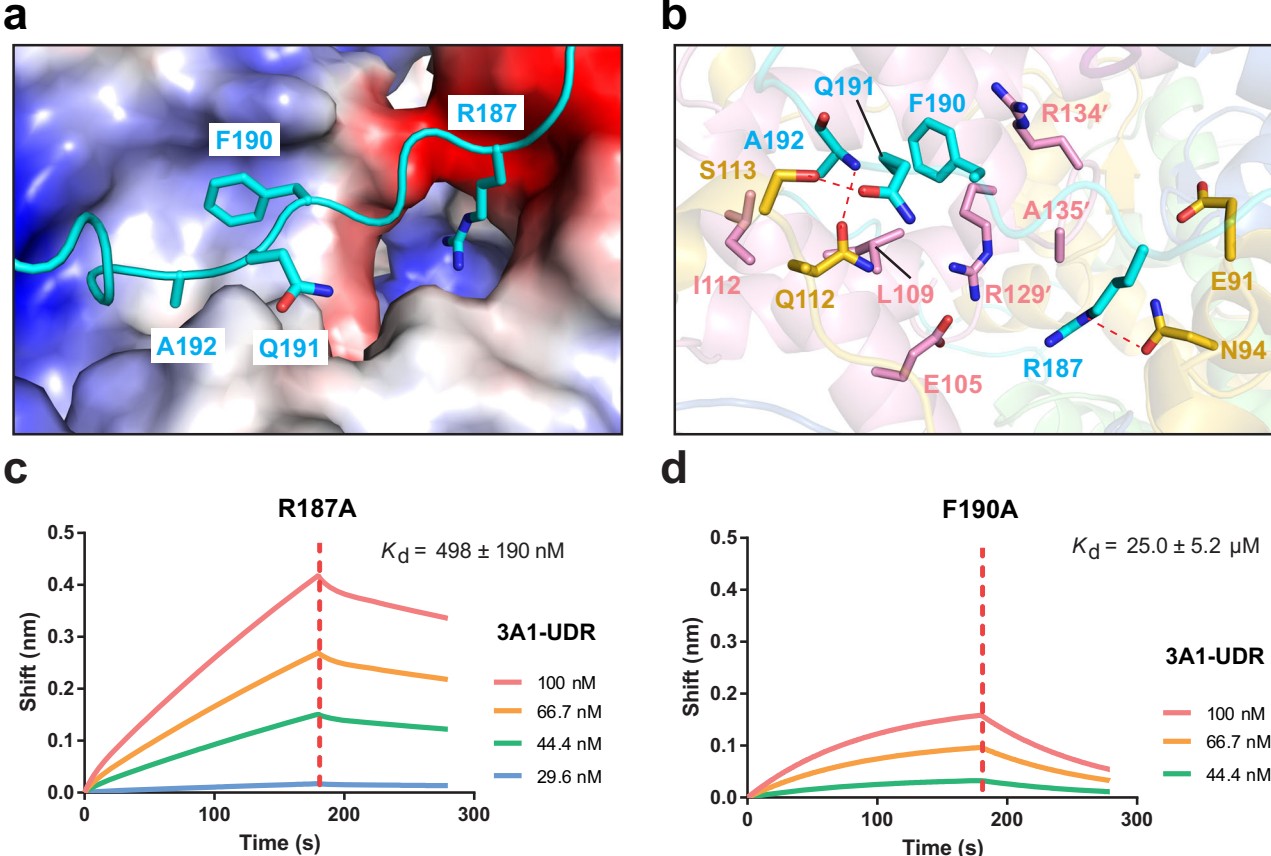

**Fig. 3 | Structural basis for the interaction between DNMT3A1 UDR and H2A C-terminal segment/H3 homodimeric interface. a** Electrostatic surface of the groove formed by H2A C-terminal segment and H3 homodimeric interface anchoring the middle segment of DNMT3A1 UDR (cyan). **b** Close-up view of the interaction of DNMT3A1 UDR (cyan) with H2A (gold) and H3 (pink) residues.

Hydrogen-bonding interactions are shown as dashed lines. Prime symbol denotes residues from the symmetry-related H3 subunit. **c**, **d** BLI assays for the interaction between R187A- (**c**) or F190A-mutated (**d**) DNMT3A1 UDR and H2AK119ub1-modified NCP. Data are mean ± s.d. ($n = 2$ biological replicates). One representative set of BLI binding data is shown.

flanked by H2A Y57, A60, E61 and E64, and the backbone carbonyl of DNMT3A M185 receives a hydrogen bond from the backbone amide of H2A E91 (Fig. 2d and Supplementary Fig. 4d). Together, these interactions underpin the contact between DNMT3A1 UDR and the H2A-H2B acidic patch.

Next, we selected key DNMT3A1 residues underlying the interaction with the H2A-H2B acidic patch for mutagenesis and subsequent BLI binding assays (Fig. 2e–h and Supplementary Fig. 6a, b). In comparison with wild-type (WT) DNMT3A1 UDR, the UDR mutants R171A, W176A and L180A all showed significantly impaired binding to the H2AK119ub1-modified NCP, with the measured affinities decreased by 193, 6.8 and 687 folds, respectively (Fig. 2e–g and Supplementary Fig. 6a, b). Furthermore, introducing the R181A mutation into DNMT3A1's UDR almost completely abolished the interaction with H2AK119ub1-modified NCP (Fig. 2h). Together, these data lend a strong support to the observed interaction between DNMT3A1 UDR and the H2A-H2B acidic patch of nucleosome.

### Structural basis for the interaction of DNMT3A1 UDR with H2A C-terminal segment and H3 homodimeric interface

The DNMT3A1 segment (residues Q182-D194) downstream of the U-turn extends along the surface groove formed by H3 homodimer and H2A C-terminal segment (α3-αC and C-terminal tail), located next to the nucleosome dyad (Figs. 1e, 3a and Supplementary Fig. 4a). Of note, the side chain of DNMT3A1 R187 inserts into the pocket formed by H2A E91 and N94, H3 E105, and R129′ and A135′ from the second H3 molecule (denoted by prime symbol) to engage in electrostatic and/or

van der Waals contacts (Fig. 3b and Supplementary Fig. 4e). The aromatic ring of DNMT3A1 F190 packs against the side chains of H3 L109 and R129′ for van der Waals contacts and H3 R134′ for a cation-π interaction (Fig. 3b and Supplementary Fig. 4f). The sidechain carbonyl of DNMT3A1 Q191 receives a hydrogen bond from the sidechain hydroxyl group of H2A S113, while the backbone amide of DNMT3A1 A192 donates a hydrogen bond to the sidechain carbonyl of H2A Q112 (Fig. 3b and Supplementary Fig. 4f). In addition, the side chain of DNMT3A1 A192 is positioned in proximity with the side chains of H3 L109 and I112 for hydrophobic contacts (Fig. 3b and Supplementary Fig. 4f).

Consistent with the structural observations, introducing the DNMT3A1 R187A and F190A mutations led to 30-fold and >1000-fold reduction, respectively, in the DNMT3A1 UDR binding to the H2AK119ub1 NCP (Fig. 3c, d and Supplementary Fig. 6a, b). Thus, the surface groove formed by H2A C-terminal segment and H3 homodimeric interface provides another platform for the nucleosome interaction with DNMT3A1's UDR.

### Structural basis for the interaction of DNMT3A1 UDR with nucleosomal DNA and H2AK119ub1

Cryo-EM density analysis of the DNMT3A1 UDR−H2AK119ub1 NCP complex reveals ~7 Å local resolution for the C-terminal fragment of DNMT3A1 UDR and H2AK119C-conjugated ubiquitin (Supplementary Fig. 3c), indicative of conformational dynamics. Nevertheless, we were able to trace a helical density corresponding to DNMT3A1 K200-W210, sandwiched between H2AK119ub1 and nucleosomal DNA at superhelix

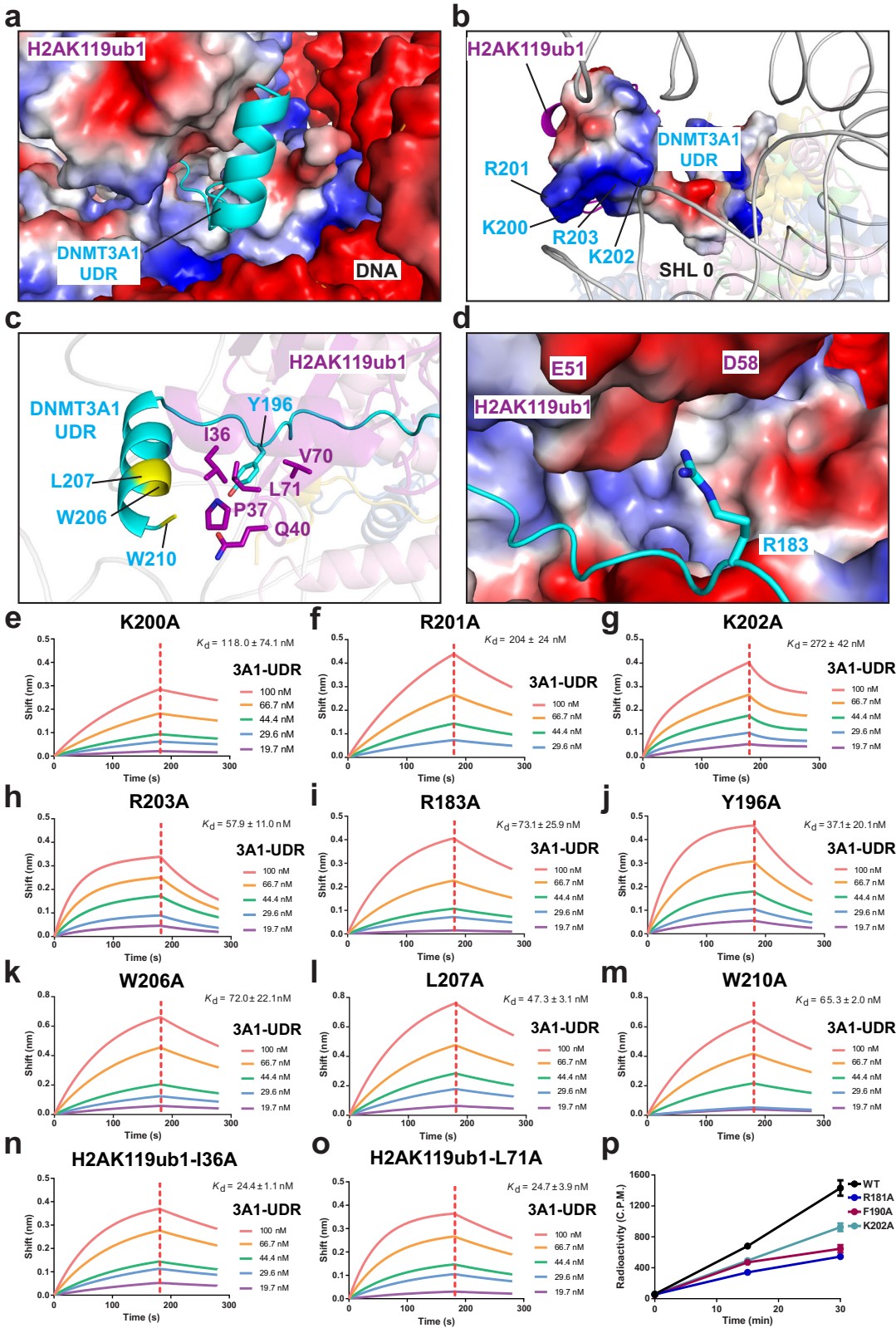

location 0 (SHL 0) (Fig. 1e and Fig. 4a). Although the side chains of the helical residues of DNMT3A1 were not traceable due to limited resolution, their proximity with ubiquitin and nucleosomal DNA provides a molecular explanation for the H2AK119ub1-dependent interaction between DNMT3A1 UDR and NCP (Fig. 4a, b), and an explanation to the previously observed mutational effects by some of the residues in this region (e.g. K200-R203)[12,28]. At the beginning of the DNMT3A1 α-helix,

a basic tetrad of DNMT3A1 (K200, R201, K202 and R203) is positioned in proximity with the SHL 0 DNA (Fig. 4b), suggesting a role of these residues in mediating the interaction of DNMT3A1 UDR with nucleosomal DNA. Residues W206, L207 and W210 from the DNMT3A1 α-helix are positioned in proximity with ubiquitin I36, P37, Q40 and L71, indicative of van der Waals intermolecular contacts (Fig. 4c). In addition, the aromatic ring of DNMT3A1 Y196 is in a position for

**Fig. 4 | Structural and biochemical analysis of the interaction of DNMT3A1 UDR with nucleosomal DNA and H2AK119ub1 and the impact of the UDR-H2AK119ub1 NCP interaction on DNMT3A1-mediated DNA methylation.** **a** Electrostatic surface of nucleosomal DNA and H2AK119ub1 bound to the C-terminal α-helix of DNMT3A1 UDR in cartoon representation. **b** Electrostatic surface view of the C-terminal α-helix of DNMT3A1 UDR, with the residues in proximity with nucleosomal DNA labeled. **c** Close-up view of the van der Waals contacts between DNMT3A1 UDR and a surface patch of H2AK119ub1 (purple). **d** DNMT3A1 R183 in proximity with H2AK119ub1 for potential electrostatic contact. The H2AK119ub1 residues near DNMT3A1 R183 are labeled. **e–m** BLI assays for the interaction between K200A- (**e**), R201A- (**f**), K202A- (**g**), R203A- (**h**), R183A- (**i**), Y196A- (**j**), W206A- (**k**), L207A- (**l**), or W210A-mutated DNMT3A1 UDR (**m**) and H2AK119ub1-modified NCP. **n, o** BLI assays for the interaction between DNMT3A1 UDR and H2AK119ub1 I36A- (**n**) or H2AK119ub1 L71A- (**o**) modified NCP. For (**e–o**), data are mean ± s.d. ($n = 2$ biological replicates). One representative set of BLI binding data is shown. (**p**) In vitro DNA methylation kinetics of full-length DNMT3A1, WT or mutant in the context of DNMT3A1-DNMT3L complex, on H2AK119ub1-modified NCP with linker NDA containing multiple CpG sites. Data are mean ± s.d. ($n = 3$ biological replicates). Source data are provided as a Source Data file.

hydrophobic contacts with the side chains of ubiquitin V70 and L71 (Fig. 4c and Supplementary Fig. 5g); and the side chain of DNMT3A1 R183 is in proximity with a surface of ubiquitin containing negatively charged residues (e.g. E51 and D58) for electrostatic attraction (Fig. 4d).

We then selected the DNA-contacting DNMT3A1 residues for mutagenesis. Consistent with a previous observation[12], the K200A, R201A, K202A and R203A mutations led to 7.1-, 12.3-, 16.4- and 3.5-fold reduction, respectively, in the DNMT3A1 UDR binding to H2AK119ub1 NCP (Fig. 4e–h and Supplementary Fig. 6a, b), confirming that the DNA contacts of these residues contribute to the DNMT3A1–NCP association. Furthermore, we performed mutational analysis of the DNMT3A1 residues that are positioned near H2AK119ub1, including R183, Y196, W206, L207 and W210. Our results show that mutating each of these residues to alanine led to a decrease in the DNMT3A1 UDR binding to H2AK119ub1 NCP by 2.2–4.4 folds (Fig. 4i–m and Supplementary Fig. 6a, b), consistent with the previous observation that a quadruple DNMT3A1 mutation, Y196A/Y197A/W206A/W210A, eliminates the H2AK119ub1-dependent nucleosomal association[12]. In addition, we performed mutational analysis for H2AK119ub1-conjugated ubiquitin residues, including I36 and L71. The ubiquitin I36A and L71A mutations each decreased the binding between DNMT3A1 UDR and H2AK119ub1 NCP by ~1.5 fold (Fig. 4n, o and Supplementary Fig. 6a, b), thereby supporting an involvement of this ubiquitin region for the interaction between DNMT3A1 UDR and H2AK119ub1 NCP. Together, these data reveal the molecular basis for the DNMT3A1 interactions with H2AK119ub1 and nucleosomal DNA.

## Impact of DNMT3A1 UDR–H2AK119ub1 NCP interaction on DNMT3A1-mediated DNA methylation in vitro

To determine how the DNMT3A1 UDR–H2AK119ub1 interaction affects DNMT3A1-mediated DNA methylation in the nucleosome context, we measured the in vitro DNA methylation kinetics of full-length DNMT3A1-DNMT3L complex (Supplementary Fig. 7) on a nucleosome substrate with one of the linker DNA regions containing multiple CpG sites. Our results indicate that introducing the H2AK119ub1 NCP-binding defective mutation, such as R181A, F190A and K202A, into full-length DNMT3A1 led to a notable decrease of the methylation rate for the nucleosome substrate, with the DNA methylation efficiency reduced by 2.6, 2.2 and 1.6 folds, respectively, after a 30-min reaction (Fig.4p). It is noted that among the three mutations, the R181A mutation impaired the DNA methylation activity of DNMT3A1 most pronouncedly, in line with the fact that this mutation led to the greatest disruption of the DNMT3A1 UDR–H2AK119ub1 NCP interaction. Together, these results suggest that the DNMT3A1 UDR–H2AK119ub1 interaction underpins DNMT3A1-mediated DNA methylation in the H2AK119ub1-enriched chromatin regions.

## Impact of UDR mutations on the activity of DNMT3A1 in cells

To evaluate potential impact of the DNMT3A1 UDR–H2AK119ub1 interaction on cellular DNA methylation, we stably transduced exogenous DNMT3A1, either WT or an UDR-defective mutant (R181A or F190A), into mouse ESCs with triple knockout of Dnmt1, Dnmt3a and Dnmt3b (TKO)[41] (Fig. 5a). Consistent with a previous report[11], our

immunofluorescence (IF) assay revealed that the ectopically expressed WT DNMT3A1 was localized exclusively in nuclei and concentrated at the DAPI-dense heterochromatic regions (Fig. 5b, c). In contrast, the F190A mutant of DNMT3A1 showed a diffused pattern in the nuclei and was significantly excluded from DAPI-dense heterochromatin (Fig. 5b, c). In addition, the DNMT3A1 F190A was also observed to be present in cytoplasm (Fig. 5b), which was further confirmed by the chromatin fractionation assays (Fig. 5d). Our observation for the cellular localization of DNMT3A1 F190A is reminiscent of what was previously reported for DNMT3A2[11], suggesting that the UDR–NCP interaction may contribute to the distinct cellular localization patterns seen for DNMT3A1 and DNMT3A2[11]. On the other hand, we did not observe appreciable defects with R181A, another NCP binding-defective UDR mutant. Compared with WT DNMT3A1, the R181A mutant exhibited a similar pattern of cellular localization (Fig. 5b, c) and strong chromatin association (Fig. 5d). Furthermore, mass spectrometry-based measurement of global DNA methylation showed that unlike the F190A mutant, R181A-mutated DNMT3A1 was able to rescue genomic DNA methylation as WT DNMT3A1 (Fig. 5e and Supplementary Data 1).

The quite strong defects seen in vitro with R181A (Figs. 2h, 3d), as well as a lack of appreciable defect seen with the same mutant in TKO cells, implies the existence of additional factors that may coordinate with the UDR for regulating DNMT3A1's functionalities. In support of this notion, previous studies have indicated that the recruitment of DNMT3A to heterochromatin is regulated by heterochromatin protein 1 (HP1) and histone lysine 9 methyltransferase SETDB1[42–44]. Alternatively, it remains possible that DNMT3A R181A mutation may lead to focal defects in DNA methylation at H2AK119ub1-positive genomic regions. How exactly the DNMT3A1 UDR orchestrates the DNMT3A1-unique functionality in cells awaits further investigation in the future.

## Structural comparison of DNMT3A1–H2AK119ub1 NCP and JARID2-H2AK119ub1 NCP interaction

In addition to DNMT3A1, previous studies have identified other H2AK119ub1 readers such as JARID2 (a cofactor of PRC2.2 complex[37,38]), RING and YY1 binding protein (RYBP, a component of variant PRC1 [vPRC1][45,46], remodeling and spacing factor (RSF1)[47], and Zuotin-Related Factor (ZRF1)[48]. Among these, the cryo-EM structure of the JARID2-containing PRC2.2 complex bound to H2AK119ub1 NCP was reported[49]. Structural analysis of the JARID2–H2AK119ub1 NCP interaction reveals a multivalent binding mechanism similar to that of DNMT3A1 UDR–H2AK119ub1 NCP. JARID2's N-terminal segment (residues 22–56) interacts with H2AK119ub1 and nucleosomal DNA through its N-terminal ubiquitin interaction motif (UIM), while it binds to the H2A-H2B acidic patch via a subsequent protein segment rich in positively charged residues (Fig. 6a, b)[49]. In fact, structural overlay of the DNMT3A1 UDR–H2AK119ub1 and JARID2–H2AK119ub1 complexes reveals great overlap between the DNMT3A1- and JARID2-occupied histone surfaces, which involve a nearly identical surface channel extending from the H2A-H2B acidic patch to the H3 homodimeric interface (Fig. 6c).

The main structural difference between the DNMT3A1 UDR–H2AK119ub1 and JARID2–H2AK119ub1 contacts lies in that the DNMT3A1 and JARID2 fragments are orientated on NCP in an opposite

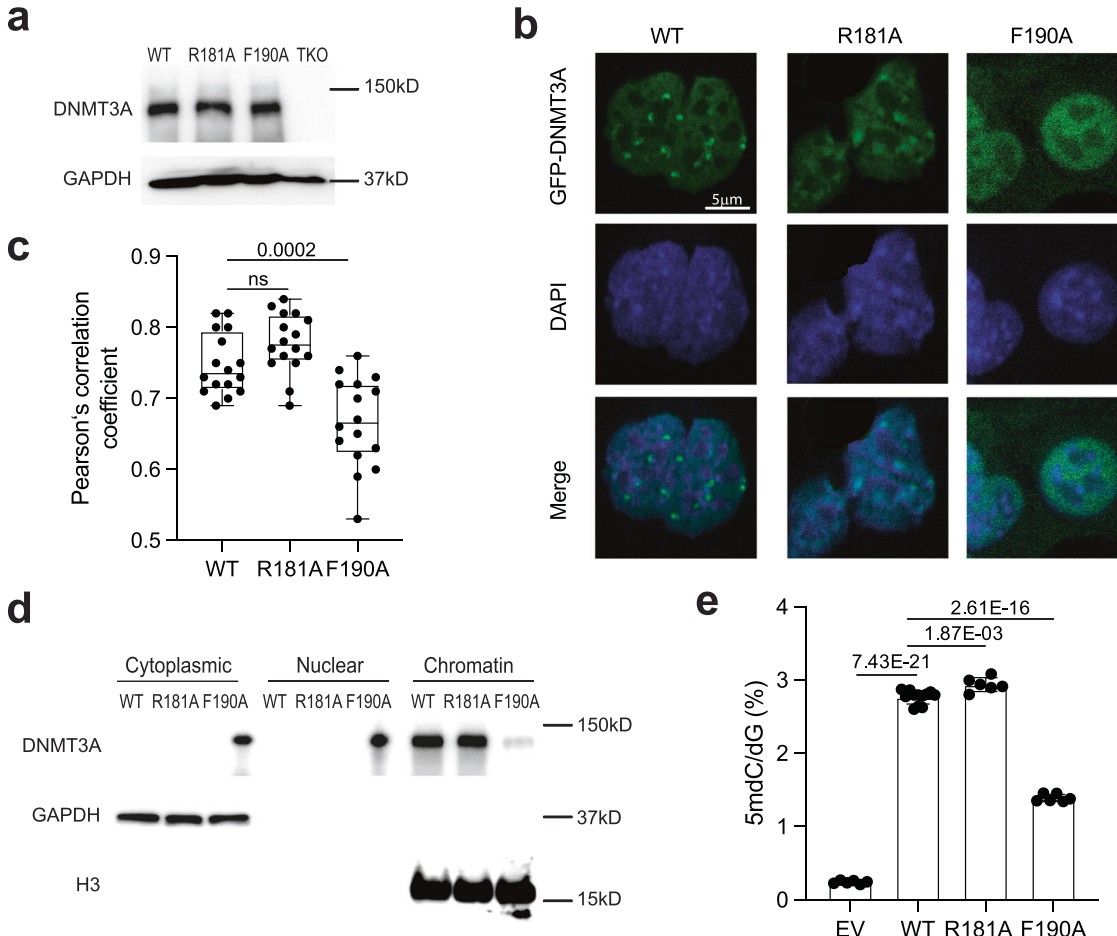

**Fig. 5 | Effect of the UDR-nucleosome interaction on the activity of DNMT3A1 in cells. a** Western blotting of the indicated DNMT3A1 in TKO cells. The molecular weight markers are labeled on the right. The experiment was performed twice with similar results. **b-c** Immunofluorescence of the indicated DNMT3A1 (anti-GFP) and DAPI (**b**) and box plot (**c**) showing correlation coefficient between DNMT3A and DAPI ($n = 20$ cells). Scale bar in (**b**), 5 μm. The whiskers in (**c**) are minimum to maximum, the box depicts the 25th–75th percentiles, and the line in the middle of the box is plotted at the median. Two-sided unpaired Student $t$-test was used for statistical analysis. $P$-values are labeled on top of each comparison. **d** Chromatin fractionation assay followed by blotting of DNMT3A1 in the indicated cell fraction. Cytoplasm, cytoplasmic fraction; Nuclear, nucleoplasmic fraction; Chromatin, chromatin-bound fraction. The experiment was performed twice with similar results. **e** Mass spectrometry-based quantification for global DNA methylation levels in TKO cells with stable expression of the indicated DNMT3A1. Two-sided unpaired Student $t$-test was used for statistical analysis. $P$-values are labeled on top of each comparison. Data are mean ± s.d. (EV: $n = 6$; WT: $n = 12$; R181A: $n = 6$; F190A: $n = 6$). Source data are provided as a Source Data file.

polarity: DNMT3A1 UDR extends from H2A-H2B acidic patch to H2AK119ub1, whereas JARID2 extends from H2AK119ub1 to H2A-H2B acidic patch (Fig. 6c). In addition, the H2AK119ub1-interacting helices of DNMT3A1 and JARID2 are positioned in discrete regions on the ubiquitin moiety (Fig. 6c): JARID2 interacts with the I44-centered hydrophobic patch via its N-terminal UIM, as typically observed for the UIM-ubiquitin interaction[50]; in contrast, the α-helix of DNMT3A, lack of a classical UIM motif, interacts with the hydrophobic patch around I36, known as alternative ubiquitin-recognition site[50]. How other H2AK119ub1 readers interact with H2AK119ub1-modified NCP remains to be determined.

## DNMT3A1 UDR and JARID2 N-terminus bind to H2AK119ub1 in a mutually exclusive manner

The fact that DNMT3A1 UDR and JARID2 occupy the same surface of H2AK119ub1-modified NCP raises a possibility that DNMT3A1 may compete with JARID2 for the binding to H2AK119ub1-NCP, indicative of a potential crosstalk between the two. To test this idea, we first performed BLI assay to measure the binding between GST-tagged N-terminal segment of JARID2 (residues 1–60, JARID2$_{1-60}$) and H2AK119ub1-modified NCP. Note that introducing the GST tag helps to ensure proper BLI signal to be obtained[51]. In contrast to GST alone that showed no appreciable binding, GST-JARID2$_{1-60}$ interacted with H2AK119ub1-modified NCP with a $K_d$ of 250 nM (Supplementary Fig. 8a, b). Next, we measured the binding of GST-tagged JARID2 fragment (residues 1–60, JARID2$_{1-60}$) to H2AK119ub1-modified NCP pre-incubated with various concentrations of DNMT3A1 UDR. Inspection of the kinetic curves reveals that increasing concentration of DNMT3A1 UDR led to a gradual decrease of response signal detected for GST-JARID2$_{1-60}$ association, reaching a ~70% decrease when the molar ratio between DNMT3A1 UDR and GST-tagged JARID2$_{1-60}$ was increased to 1:1 (Fig. 6d). Conversely, pre-incubation of H2AK119ub1-modified NCP with increasing concentration of tag-free JARID2$_{1-60}$ also led to a gradual decrease of the response for DNMT3A1 UDR association, reaching a ~40% decrease when the molar ratio of JARID2$_{1-60}$:DNMT3A1 UDR was increased to 32:1 (Fig. 6e). These observations confirm that the DNMT3A1 UDR and JARID2$_{1-60}$ interact with H2AK119ub1-modified NCP in a mutually exclusive manner.

Together, our structural and biochemical analyses reveal a competition between DNMT3A1 UDR and JARID2$_{1-60}$ for binding to H2AK119ub1-modified nucleosome. Given the fact that JARID2 constitutes a key cofactor for the PRC2.2 complex, a writer of H3K27me3[52],

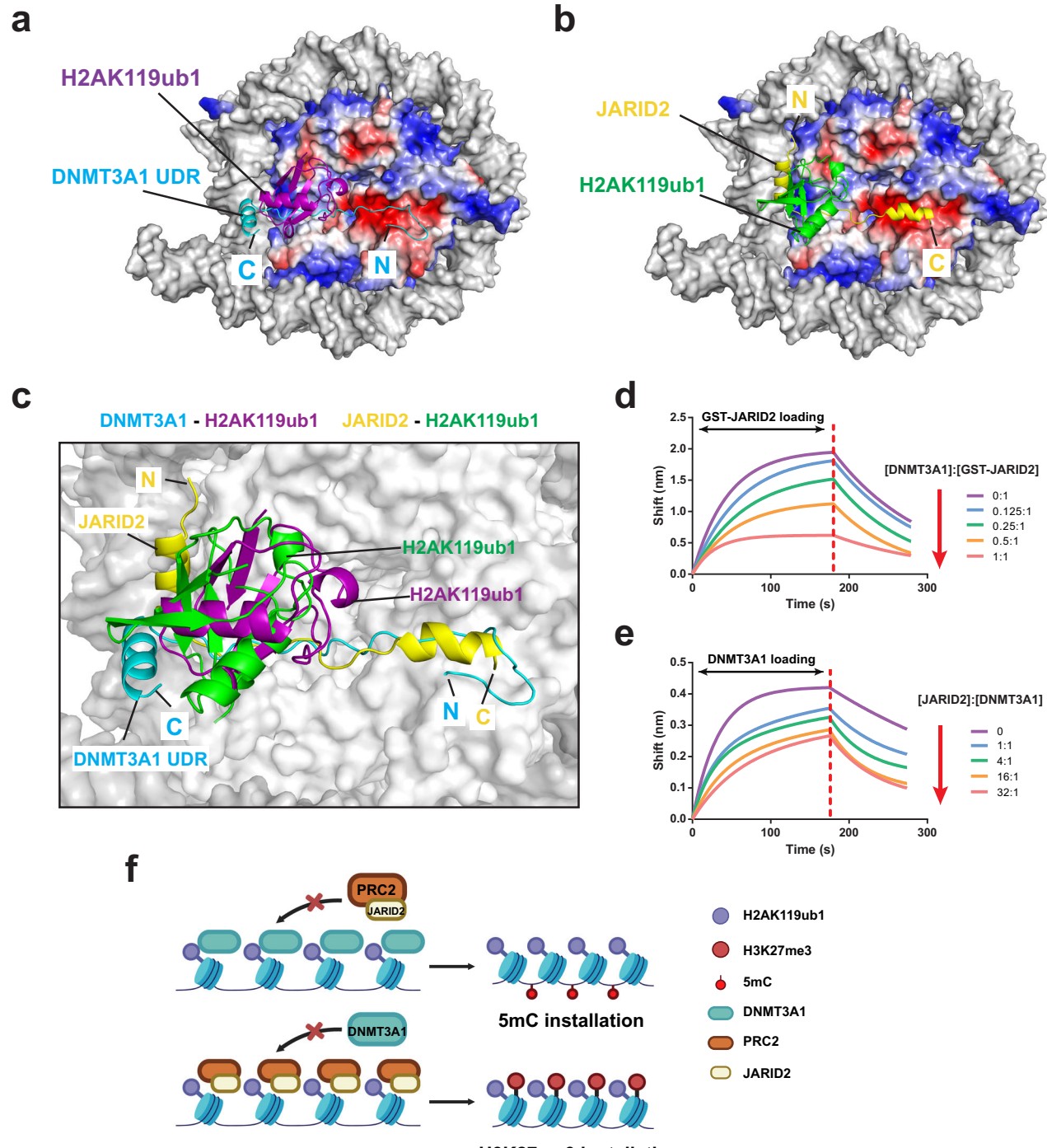

**Fig. 6 | DNMT3A1 UDR and JARID2 compete on binding to H2AK119ub1-modified NCP. a**, **b** Surface representation of H2AK119ub1-modified NCP, with DNA in gray and histones in electrostatic view, bound to DNMT3A1 UDR (**a**) or JARID2 N-terminal segment (PDB 6WKR) (**b**). The N- and C-termini of DNMT3A1 UDR and JARID2 segments are labeled with letters 'N' and 'C', respectively. **c** Structural overlay of the DNMT3A1 UDR–H2AK119ub1 NCP and JARID2–H2AK119ub1 NCP (PDB 6WKR) complexes. DNMT3A1 UDR and bound H2AK119ub1 are colored in cyan and purple, respectively. JARID2 N-terminal segment and bound H2AK119ub1 are colored yellow and green, respectively. The DNMT3A1-bound NCP is shown in surface representation. The JARID2-bound NCP was removed for clarity. **d** BLI analysis of JARID2$_{1-60}$ loading to biosensors pre-loaded with H2AK119ub1-modified

NCP and various concentrations of DNMT3A1 UDR. The various DNMT3A1:JARID2 molar ratios are indicated on the right. The change of response with increasing concentration of DNMT3A1 UDR is indicated by red arrow. **e** BLI analysis of DNMT3A1 UDR loading to biosensors pre-loaded with H2AK119ub1-modified NCP and various concentrations of JARID2$_{1-60}$ segment. The various JARID2:DNMT3A1 molar ratios are indicated on the right. The change of response with increasing concentration of JARID2 is indicated by red arrow. **f** Model for the competitive H2AK119ub1 NCP-binding between DNMT3A1 and JARID2-containing PRC2 complex, which might contribute to the mutually exclusive installation of DNA methylation (5mC) and H3K27me3.

it is conceivable that this binding competition provides a mechanism for a mutually exclusive action between DNMT3A-mediated DNA methylation and PRC2.2-mediated H3K27me3 installation on the H2AK119ub1-modified nucleosomes (Fig. 6f).

## Discussion

Proper establishment and maintenance of the DNA methylation landscape in organismal development depends on the precise regulation of DNA methyltransferases (DNMTs). Increasing evidence has indicated that direct readout of various histone modifications by functional modules within DNMTs plays a key role in ensuring the site-specific DNA methylation across the genome[17,53]. This study, through structural and biochemical characterization of the interaction between DNMT3A1 UDR and H2AK119ub1-modified NCP, provides critical mechanistic insights into the functional regulation of DNMT3A1 by H2K119ub1-marked nucleosome, and sheds light onto the functional crosstalk between two major epigenetic mechanisms: DNA methylation and polycomb signaling.

First of all, this study provides a molecular view as for how DNMT3A1 engages the H2AK119ub1-modified chromatin. Our structural analysis of the complex between DNMT3A1 UDR and H2AK119ub1-modified NCP identified DNMT3A1 residues 170–210 as a H2AK119ub1 NCP-interacting segment. Intriguingly, DNMT3A1 UDR targets a rather extensive area of histone surface, including the H2A-H2B acidic patch, H2A C-terminal segment, H3 homodimeric interface, and H2AK119ub1. The N-terminal segment of DNMT3A1 UDR forms a sharp U-turn, anchored to the H2A-H2B acidic patch via Arg-mediated cavity insertions and hydrophobic contacts by bulky aromatic and aliphatic residues. Furthermore, the DNMT3A1 segment downstream of the U-turn extends into the surface groove formed by H2A's C-terminal segment and H3's homodimeric interface, again engaging in Arg-mediated cavity insertion and aromatic residue-mediated hydrophobic contacts. Finally, the C-terminal segment of DNMT3A1 UDR forms an α-helix that contacts both H2AK119ub1 and nucleosomal DNA. Together, such a multivalent interaction underpins a strong association between DNMT3A1 and H2AK119ub1-marked nucleosome. It is worth mentioning that a previous study reported that the isoform 3 of DNMT3B (DNMT3B3) complexed with DNMT3A2 also interacts with the H2A-H2B acidic patch via a two-arginine anchoring mechanism[13] (Supplementary Fig. 9a–c), highlighting that targeting the acidic patch serves as a recurrent mechanism for chromatin recruitment of DNMT3s. However, unlike the DNMT3A1 UDR, which engages multiple histone surfaces with a buried surface area of 1162 Å² (Supplementary Fig. 9b), the reported DNMT3B3-nucleosome interaction[13] is mainly limited to the H2A-H2B acidic patch, resulting in a much reduced buried surface area (709 Å²) (Supplementary Fig. 9a). The functional implication of such distinct nucleosome contacts by different DNMT3 complexes awaits further investigation. Nevertheless, it is conceivable that DNMT3A1 is subject to functional regulation by both DNMT3B3 and the N-terminal UDR. In this context, the multivalent DNMT3A1-nucleosome interaction, in conjunction with the oligomeric assembly of DNMT3A[54,55], may contribute to the preferential association of DNMT3A1, but not DNMTA2, with the nucleosome-dense heterochromatic regions[11,14]. In support of this notion, our cellular analysis revealed that disruption of the UDR–nucleosome interaction by the F190A mutation led to the severely impaired chromatin association and DNA methylation by DNMT3A1 in mouse TKO ESCs. Interestingly, we did not observe that R181A, another UDR mutation showing strong defects in the in vitro nucleosome interaction and methylation assays, significantly affected the chromatin association and DNA methylation by DNMT3A1 in TKO cells, suggesting the existence of an unknown mechanism in functional regulation of DNMT3A1. Also, it is possible that the R181A mutation of DNMT3A1 might cause defective 5mC deposition focally at H2AK119ub1-targeted regions. Further investigation is merited to

determine whether and how DNMT3A1 interplays with various interfaces of H2AK119ub1-containing NCP and/or other unknown chromatin players, in order to orchestrate DNA methylation patterning in cells.

The H2A-H2B acidic patch has been identified as a hotspot interaction site for nucleosome-associated proteins[40]. Whereas the acidic patch-interacting elements are structurally diverse, a recurrent theme is the use of two or more arginine residues to interact with the surface cavities of the acidic patch[40]. In particular, a highly conserved interaction involves insertion of an arginine side chain into a surface cavity formed by the α2 and α3 helices of H2A and the C-terminal helix of H2B, known as canonical arginine anchor site[40]. In addition, some other arginine anchor sites, such as the shallow cleft (involving H2A E61, E64, and L65) located next to the canonical arginine anchor site serves as variant arginine anchor site to support the acidic patch recognition[40,56]. Here, DNMT3A1 UDR forms a sharp U-turn to bring two arginine residues (R171 and R181) into proximity, inserting residue R181 into the canonical arginine anchor site while residue R171 into an adjacent surface groove known as variant arginine anchor site. In addition, residue W176 inserts into a third surface cavity of the H2A-H2B acidic patch to reinforce the association. This multi-pronged interaction mediated by the U-turn fold of DNMT3A1 adds another example for H2A-H2B acidic patch-mediated protein interaction.

Second, this study reveals an intricate interplay between DNMT3A1-mediated DNA methylation and polycomb signaling. The DNMT3A1 UDR−H2AK119ub1 interaction appears important for proper regulation of bivalent neurodevelopmental genes[12,28]. Homozygous knockout of DNMT3A1, but not DNMT3A2, led to postnatal mortality of mouse[12]. Through the in vitro DNA methylation analysis of full-length DNMT3A1−DNMT3L complex on nucleosome substrates, this study demonstrated that disruption of the DNMT3A1 UDR−H2AK119ub1 NCP interaction affects the DNA methylation activity of the DNMT3A1-DNMT3L complex, thereby providing a direct link between the DNMT3A1 UDR−H2AK119ub1 interaction and DNMT3A1-mediated DNA methylation in H2AK119ub1-enriched chromatin regions.

Furthermore, our comparative structural analysis reveals that DNMT3A1 UDR and PRC2.2 cofactor JARID2 bind to the same surface channel of H2AK119ub1-modified NCP, formed by the H2A-H2B acidic patch, H2A C-terminal segment and H3 homodimeric interface. This observation, together with our in vitro competitive nucleosome binding assays, suggests that DNMT3A1 and PRC2.2 complex may bind to H2AK119ub1-modified nucleosome in a mutually exclusive fashion. H2AK119ub1 is a prevalent histone modification that plays a crucial role in recruiting PRC2.2 subcomplex for deposition of repressive H3K27me3 mark[34,52,57]. DNA methylation and H3K27me3 represent two different epigenetic mechanisms that mediate gene repression[58]. Whereas the co-existence of these two marks has been observed in some loci[59], they were reported to be mutually exclusive in many other loci, such as CGIs[60–63]. In fact, most of the polycomb target genes remain unmethylated throughout development[35]; for regions enriched with DNA methylation, removal of DNA methylation triggers the expansion of H3K27me3 from H3K4me3/H3K27me3 bivalent promoters[62]. The underlying mechanism for the mutual antagonism between DNA methylation and H3K27me3 may in part be attributed to the crosstalk between polycomb complex and DNA methylation machinery. For instance, it was shown that DNMT3A/DNMT3B regulatory protein DNMT3L[64,65] antagonizes DNMT3A/DNMT3B-mediated DNA methylation at the H3K27me3-enriched regions via a competing interaction with PRC2 complex in embryonic stem cells[66]. Furthermore, the H3K36me2/3 mark was shown to recruit DNMT3A/DNMT3B via an interaction with their PWWP domains[27,67,68], while enzymatically inhibit PRC2-mediated H3K27 methylation[69,70]. This study reveals that the N-terminal domain of DNMT3A1 and the PRC2 cofactor JARID2 interact with H2AK119ub1-modified NCP in a mutually exclusive manner, providing another mechanism that may contribute to the competing presence between DNA methylation and H3K27me3.

We note limitations of this study. While this study delineates an extensive interaction of DNMT3A1 UDR with histone surface, including the H2A-H2B acidic patch, the modest local resolution of the DNMT3A1 UDR and H2AK119ub1 prevented us from a more thorough characterization of the UDR-mediated interaction, such as the UDR–H2AK119ub1 contact. Second, DNMT3A regulation is complex. While extensive structural and biochemical studies provide critical glimpse of DNMT3A1's functionalities, detailed investigation of those functional mutants using full-length DNMT3A1 under appropriate cellular contexts is required in the future.

## Methods

### Expression and purification of DNMT3A1 and JARID2 proteins

The gene fragment encoding residues 126–223 of human DNMT3A1 (a.k.a. DNMT3A1 UDR herein) was PCR amplified and inserted into a modified pRSFDuet-1 vector, preceded by a $His_6$-SUMO tag and Ubl-specific protease (ULP) 1 cleavage site. The $His_6$-SUMO-DNMT3A1 UDR fusion protein was expressed in *Escherichia coli* BL21 DE3 (RIL) cells, followed by purification using a $Ni^{2+}$-NTA column. Subsequently, the $His_6$-SUMO tag was removed by ULP1 cleavage and the DNMT3A1 UDR protein was sequentially purified via ion-exchange chromatography on a Heparin column (GE Healthcare), nickel affinity chromatography on a $Ni^{2+}$-NTA column and size-exclusion chromatography on a HiLoad 16/600 Superdex-75 pg column pre-equilibrated with buffer containing 20 mM HEPES (pH 7.5), 50 mM NaCl and 5 mM dithiothreitol (DTT). The purified DNMT3A1 UDR sample was concentrated and flash-frozen in liquid nitrogen and stored at −80 °C before use. DNMT3A1 UDR mutants were generated using site-directed mutagenesis and purified in the same way as that for WT DNMT3A1 UDR.

For the JARID2–H2AK119ub1 binding assay, the DNA fragment encoding residues 1–60 of human JARID2 (referred to hereafter as $JARID2_{1-60}$) was synthesized by Integrated DNA Technologies and inserted into pRSFDuet-1 (for production of tag-free $JARID2_{1-60}$) or pGEX-6P-1 (for production of GST-$JARID2_{1-60}$) vector. Tag-free $JARID2_{1-60}$ was purified using the same method as DNMT3A1 UDR, except that no ion-exchange chromatography was involved and the buffer for size-exclusion chromatography contained 500 mM instead of 50 mM NaCl. GST-tagged $JARID2_{1-60}$ was purified via GST affinity chromatography, ion exchange and size-exclusion chromatography. The purified GST-tagged $JARID2_{1-60}$ sample was stored in buffer containing 20 mM HEPES (pH 7.5), 500 mM NaCl and 5 mM DTT at −80 °C before use. The free GST tag used for the BLI assay was obtained from the purified GST-$JARID2_{1-60}$ after cleavage by preScission protease, a second GST affinity chromatography, and size-exclusion chromatography on a HiLoad 16/600 Superdex-75 pg column pre-equilibrated with buffer containing 50 mM Tris-HCl (pH 8.0) and 300 mM NaCl. The protein sample was concentrated and stored at −80 °C before use.

For DNA methylation assays, full-length DNMT3A1 and DNMT3L were each cloned into an in-house MBP-tagged vector and expressed in *E. coli* BL21 DE3 (RIL) cells. The DNMT3A1- and DNMT3L-expressed cells were then co-lysed and the DNMT3A1-DNMT3L complex was purified using the same method as DNMT3A1 UDR, except that the MBP tag was removed by Tobacco Etch Virus (TEV) protease, followed by size-exclusion chromatography on a Superdex 200 increase 10/300 GL column in buffer containing 20 mM HEPES (pH 7.5), 5 mM DTT, and 250 mM NaCl. Full-length DNMT3A1 mutants were generated using site-directed mutagenesis and produced in the same approach as that for WT protein.

### Expression and purification of histone proteins and H2AK119ub1 modification

Preparation of *Xenopus* H3, H4, H2A and H2B followed previously published protocols[6,71] except that H2A and H2B were cloned in tandem into the pRSFDuet-1 vector and expressed in a soluble form using *E. coli* BL21 DE3 (RIL) cells. The H2A and H2B proteins were then co-

purified using a $Ni^{2+}$-NTA column, followed by ULP1 treatment for removal of $His_6$-SUMO tag and ion-exchange chromatography on a Heparin column (GE Healthcare). $His_6$-SUMO-tagged ubiquitin G76C was inserted into pRSFDuet-1 vector, expressed in *E. coli* BL21 DE3 (RIL) cells and purified using $Ni^{2+}$-NTA column. $His_6$-SUMO-tagged ubiquitin I36A/G76C or L71A/G76C mutant was generated using site-directed mutagenesis and purified in the same way as that for $His_6$-SUMO-tagged ubiquitin G76C. The H2AK119ub1 modification was generated through conjugating G76C-mutated ubiquitin with histone H2AK119C via dichloroacetone crosslinker. In essence, $His_6$-SUMO-tagged ubiquitin G76C, I36A/G76C or L71A/G76C mutant and H2AK119C-H2B were mixed in 6:1 molar ratio in 20 mM Tris-HCl (pH 7.5), 600 mM NaCl, 5% Glycerol and 5 mM tris (2-carboxyethyl) phosphine (TCEP), followed by addition of crosslinker 1,3-dichloroacetone with the amount equal to one-half of the total sulfhydryl groups. The reaction continued overnight, before being quenched by 5 mM β-mercaptoethanol (ME), which was then removed via dialysis against buffer containing 20 mM Tris-HCl (pH 7.5), 600 mM NaCl and 5% Glycerol. Next, the H2AK119C-ubiquitin conjugate (denoted as H2AK119ub1 herein)-H2B sample was sequentially purified through $Ni^{2+}$-NTA affinity chromatography, ion-exchange chromatography on a Heparin column (GE Healthcare), removal of $His_6$-SUMO tag via ULP1 cleavage, and a second round of $Ni^{2+}$-NTA affinity chromatography. The purified H2AK119ub1-H2B protein sample was concentrated and stored at −80 °C before use.

### Reconstitution of nucleosome core particles (NCP)

The 601 nucleosomal DNA was generated by PCR amplification (forward primer F1: 5′-GCTCTCTACGTAAACATCCTGGAGAATCCCGGTGC-3′; reverse primer R1: 5′-CGAAGTGGGTAAGTCACAGGATGTATATA TCTGACACG-3′) of the 147-bp 601 DNA[72] and purified using a Mono-Q column (GE Healthcare). For Biolayer Interferometry (BLI) assay, the 601 DNA was also generated by PCR using the forward primer containing a biotin tag at the 5′ end (forward primer F2: 5′-biotin-CTCTCTCCGTAAACATGCTGGAGA-3′; reverse primer R1). For in vitro DNA methylation assay, the nucelosomal DNA was generated by PCR using the forward primer containing multiple CpG sites (forward primer F3: 5′-GCATGCGCCGTCGTTAAGCGCCCCGTGTCGAGAATCCCGG TGCCGAGG-3′; reverse primer R1).

The histone H2A-H2B tetramer, H3 and H4 were mixed in a ratio of 1.1:1:1:1 and dialyzed against buffer containing 20 mM Tris-HCl (pH 7.5), 7 M Urea at 4 °C overnight, followed by dialysis against buffer I containing 20 mM Tris-HCl (pH 7.5), 7 M Guanidine-HCl and 10 mM DTT for 3 hr, buffer II containing 20 mM Tris-HCl (pH 7.5), 2 M NaCl and 5 mM β-ME for 4 hr, and buffer II with additional 1 mM EDTA overnight. After concentration, the histone octamer was further purified using a HiLoad 16/600 Superdex-200 pg column (GE Healthcare). The purified histone octamer sample was concentrated and stored at −80 °C before use.

The histone octamer and nucleosomal DNA were mixed in a molar ratio of 0.9:1, and further mixed with 4 M KCl in equal volume. A salt gradient method was used for histone octamer reconstitution with nucleosome DNA. In essence, the samples were first dialyzed against the refolding buffer containing 10 mM Tris-HCl (pH 7.5), 2 M KCl, 1 mM DTT, and 1 mM EDTA. The concentration of KCl was then gradually decreased from 2 M to 250 mM over a period of 18 hr. Finally, the samples were dialyzed towards buffer containing 10 mM HEPES (pH 7.5) and 1 mM DTT at 4 °C overnight. The reconstituted nucleosomes were kept at 4 °C before use.

### Assembly of the complex between DNMT3A1 UDR and H2AK119ub1-modified nucleosome

The H2AK119ub1-modified nucleosome and DNMT3A1 UDR were mixed at a molar ratio of 1:8, and incubated on ice for 30 min. The sample was subject to chemical crosslinking via GraFix method[73]. An

amount of 200 pmol of sample was loaded onto the top of a 10–30% linear glycerol gradient in a buffer (50 mM HEPES, pH 7.5, 50 mM NaCl, 30% Glycerol) with a gradient of 0–0.05% (v/v) of crosslinker glutaraldehyde in each ultracentrifuge tube for GraFix, followed by centrifugation at 40,000 × g for 18 hr using a Beckman SW41 rotor at 4 °C. Then the samples were separated from top to bottom at a volume of 500 μL each for 25 fractions. The fractions were examined by 6% native PAGE gel and negative-staining electron microscopy (EM). Fractions with good homogeneity and reasonable particle size were pooled and buffer exchanged to 10 mM Tris-HCl (pH 7.5), 50 mM NaCl, 5% Glycerol and 1 mM DTT using a PD10 column (Cytiva). The samples were concentrated and stored on ice before use.

## Cryo-EM sample preparation and data collection
The complex of DNMT3A1 UDR with H2AK119ub1-modified NCP was adjusted to a concentration of 3.5 μM. A volume of 4 μL of the complex sample was applied on Copper 300 mesh grids (Quantifoil R 1.2/1.3 100 holey carbon films) after fresh glow discharge at 20 mA for 1 min. Subsequently, the sample on grids was vitrified and plunge-frozen in liquid ethane using Vitrobot (thermos scientific) with the chamber equilibrated at 100% humidity, 6.5 °C. The blot time, wait time and drain time were 4 s, 10 s and 0 s respectively and a blot force of 1 was used.

The cryo-EM data collection for the complex between DNMT3A1 UDR and H2AK119ub1-modified NCP was performed on a Titan Krios microscope operated at 300 kV at Nation Center for CryoEM Access and Training (NCCAT). Micrographs were collected in counting mode with an energy filter (slit width of 10–20 eV) at an apoF-calibrated pixel size of 0.926 Å. The exposures were recorded with an accumulated total dose of 50 e/Å$^2$ over 40 frames and a defocus range of −0.8 μm to −2.5 μm.

## Cryo-EM image processing
The cryo-EM data were processed using cryoSPARC (v4.0.1)[74]. The multi-frame movies were motion-corrected and dose-weighted using the patched motion correction module. Initial particle picking was performed using the TOPAZ method[75], resulting in a total of 3.5 million particles extracted from 9467 micrographs, with a downscaled pixel size of 2.6 Å. After several rounds of 2D classifications, 1.60 million particles from classes with discernible features of nucleosomal DNA and histone octamer were selected for further analysis. The initial models were created using 3D ab initio reconstruction and subject to heterogeneous refinement with C1 symmetry being applied. The particles associated with the class exhibiting the most clearly defined DNMT3A segment and H2AK119ub1 were re-extracted with a pixel size of 1.23 Å, followed by removal of duplicates and non-uniform refinement. Using a mask protecting density around the ubiquitin and adjacent DNA segment, particle subtraction and subsequent alignment-free 3D classification were performed using Relion 4.1[76]. Two classes with relatively strong ubiquitin density were combined and subject to another round of focused classification. Finally, 86,685 particles from the class with traceable density for both ubiquitin and DNMT3A segment were selected for final non-uniform refinement, local CTF correction and focused refinement of the ubiquitin moiety. A composite map was generated by Phenix software package (v1.20.1)[77] using combined densities from the consensus map and the local refinement map for ubiquitin.

## Model building and refinement
For model building, coordinates extracted from PDB entry 6WKR were used as a template for nucleosome and coordinates of a ubiquitin molecule predicted by AlphaFold[78] were used as a template for ubiquitin. The templates of nucleosome and ubiquitin were then fit into the overall and local refinement cryo-EM maps, respectively, using ChimeraX (v1.5)[79]. Subsequent model building of all components was

performed using Coot (v0.9.6)[80], followed by real-space refinement over the composite map using Phenix. The reported resolution was based on the gold-standard Fourier shell correlation curve (FSC) at 0.143 criterion.

## Biolayer Interferometry (BLI) assay
Binding affinities for DNMT3A, WT or mutants, GST tag, and GST-JARID2$_{1-60}$ with nucleosomes were measured on a Gator BLI platform instrument using streptavidin (SA) XT biosensors. All steps were performed at 30 °C, 1000 rpm. Reagents were formulated in the buffer containing 20 mM Tris-HCl (pH 7.5), 50 mM NaCl, 1 mM DTT and 0.01% Tween-20. The biosensors were equilibrated in the buffer for 10 min. Biotin-labeled nucleosomes were immobilized on the biosensors, followed by washing for 2 min. Then the nucleosome-loaded biosensors were submerged in solutions of DNMT3A WT or mutants, GST tag, or GST-JARID2$_{1-60}$, for 3 min and transferred to the buffer for 6 min to measure the association and dissociation kinetics. Data was analyzed on GatorOne software (v2.10.4) and results were plotted in GraphPad Prism v6.01.

## In vitro DNA methylation assay
In vitro DNA methylation assays were carried out for full-length DNMT3A1 (WT or mutant)–DNMT3L complex on H2AK119ub1-modified NCP with one of the linker DNAs containing multiple CpG sites, as described above. A 20-μL reaction mixture contained 0.2 μM H2AK119ub1-modified NCP, 0.1 μM DNMT3A1–DNMT3L tetramer, 2 μM S-adenosyl-L-[methyl-$^3$H] ($^3$H-SAM) in reaction buffer containing 50 mM Tris-HCl (pH 8.0), 100 mM NaCl, 0.05% β-ME, 5% glycerol and 200 μg/mL BSA. The reaction mixtures were incubated at 37 °C for 0, 15 or 30 min before being quenched. Ten μL of each reaction mixture was loaded onto the positively charged Nylon transfer membrane and air dried, followed by sequential washes by 0.2 M ammonium bicarbonate, Milli Q water and ethanol. After being aired-dried again, the Nylon transfer membrane was transferred to scintillation vials filled with the counting cocktail (RPI) and subject to radioactivity detection using a Beckman LS6500 counter for the tritium radioactivity. For control, the methylation assay included samples containing nucleosome and $^3$H-SAM only in the reaction buffer, which gave basal radioactivity <70 counts-per-minute (C.P.M.). The source data are provided as a Source Data file.

## Electrophoretic mobility shift assay (EMSA)
DNMT3A1 UDR sample was incubated with nucleosomes for 20 min at 4 °C in a binding buffer (pH 7.5, 50 mM NaCl, 5% Glycerol, 0.05% β-ME) and run on a native 6% polyacrylamide gel. The electrophoresis was performed using a Tris-borate buffer (25 mM Tris, 12.5 mM boric acid, pH 8.8) at a constant voltage at 100 V for 2 hr at 4 °C. Nucleosomal DNA was detected by staining with SYBR Gold and visualized by gel imager.

## Establishment of stable cell lines
The murine DNMT-TKO ESCs were transfected with the pPyCAGIZ vector (a kind gift of J. Wang, Columbia University) carrying GFP-tagged DNMT3A1, either WT or mutant, using a transfection agent of linear polyethylenimine (PEI, Sigma). Two days post-transfection, cells were subject to drug selection in the medium with 100 μg/mL zeocin (Invitrogen) for at least 10 days, followed by immunoblotting to verify DNMT3A1 expression using anti-DNMT3A antibody (abcam, ab2850).

## Chromatin fractionation assay
Chromatin fractionation assay was performed as described previously[81]. In brief, 5 million of cells were harvested and resuspended in 300 μL of ice-cold buffer A (10 mM HEPES [pH 7.9], 10 mM KCl, 1.5 mM MgCl$_2$, 0.34 M sucrose, 10% glycerol, 1 mM DTT, and 0.1% Triton X-100, freshly added with the cocktail of protease inhibitors), and incubated for 10 min on ice. After incubation, samples were

centrifuged at 1,300 g for 5 min at 4 °C. An aliquot (30 μL) of the supernatant was taken as the cytoplasmic fraction. The pellet was then washed with 500 μL of buffer A and then resuspended in 30 μL of volume buffer A. 10 x volume of buffer B (3 mM EDTA, 0.2 mM EGTA, 1 mM DTT, and protease inhibitors) was added to each sample. After brief vortex, samples were incubated for 30 min on ice and centrifuged at 1,700 g for 5 min at 4 °C. An aliquot (30 μL) of the supernatant was taken as the nucleoplasmic fraction. Pellets were washed in 500 μL of buffer B and then resuspended in 30 μL of SDS lysis buffer (50 mM Tris-Cl [pH 7.5], 2 mM EDTA, 2% SDS), which represents the chromatin fraction (fraction Chr). All fractions were then mixed with an equal amount of SDS sample loading buffer, boiled for 10 min and subject for western blotting. DNMT3A, histone H3 and GAPDH were detected using anti-DNMT3A (abcam, ab2850), anti-H3 (Cell Signaling Technology, 4499 S) and anti-GAPDH (Cell Signaling Technology, 2118 L) antibodies, respectively. The source data are provided as a Source Data file.

### Immunofluorescence (IF)
IF assay was performed as described previously[82]. Briefly, cells were fixed in 4% of paraformaldehyde for 10 min at room temperature, followed by incubation in 1 × PBS containing 0.2% of Triton X-100 for 10 min to permeabilize the cells. Fixed cells were stained with primary antibody. Anti-GFP antibody (abcam, ab290) was used for GFP-tagged DNMT3A Isoform 1 (DNMT3A1), followed by staining with the anti-rabbit Alexa-488 conjugated secondary antibody (Invitrogen, #A-11008). Nuclei were finally stained with 4,6-diamino-2-phenylindole (DAPI, 0.1 g/ml, Invitrogen, D3571). Images were taken with an FV1000 confocal microscope (Olympus; available at UNC Imaging Core). Signal colocalization analyses were performed using EzColocalization plugin of FIJI49. 20 cells of each sample were used for IF signal colocalization analysis of DNMT3A1 and DAPI. The source data are provided as a Source Data file.

### Quantifications of the levels of 5-methyl-2′-deoxycytidine (5-mdC) in genomic DNA by liquid chromatography-tandem mass spectrometry (LC-MS/MS)
The levels of 5 mdC in genomic DNA were measured as previously described[24]. Briefly, 500 ng of genomic DNA was digested with 0.1 unit of nuclease P1 (Sigma) and 0.4 unit of turbo DNase (Thermo Fisher Scientific) in a buffer containing 30 mM sodium acetate (pH 5.6) and 1 mM $ZnCl_2$ at 37 °C for 24 h. The mixture was then treated with 1 unit of Quick CIP (New England Biolabs) and 0.002 unit of phosphodiesterase I (Sigma) in 500 mM Tris-HCl (pH 8.9) at 37 °C for another 4 h. The solution was subsequently neutralized with 1 M formic acid. To 1/10 volume of the resulting nucleoside mixture was added with 0.6 pmol of $[^{13}C_5]$-5-mdC and 16.2 pmol of uniformly $^{15}N$-labeled 2′-deoxyguanosine (dG). Enzymes were then removed from the digestion mixture via chloroform extraction, the aqueous layer was dried *in vacuo*, and the dried residues were reconstition in 20 μL of doubly distilled water for LC-MS/MS analysis.

LC-MS/MS experiments were conducted using a TSQ-Altis triple-quadrupole mass spectrometer (Thermo Fisher Scientific, San Jose, CA) equipped with a Nanospray Flex source and coupled with a Dionex UltiMate 3000 UPLC for separation (Thermo Fisher Scientific, San Jose, CA). The samples were loaded onto a μ-precolumn (C18 PepMap 100, 5 μm in particle size, 100 Å in pore size, Thermo Fisher) at a flow rate of 2 μL/min within 8.5 min and then eluted onto an in-house packed Zorbax SB-C18 analytical column (5 μm in particle size, 200 Å in pore size, Michrom BioResource, Auburn, CA, 75 μm × 20 cm) at a flow rate of 300 nL/min. Formic acid (0.1%, v/v) in water and acetonitrile were used as solution A and B, respectively. A gradient of 0–95% B in 20 min, and 95% B for 10 min was employed. The TSQ-Altis triple-quadrupole mass spectrometer was operated in the positive-ion mode. The voltage

for electrospray, capillary temperature, collision energies were 2.0 kV, 325 °C and 20 V. Q1 and Q3 resolutions were 0.7 and 0.4 Th full-width at half-maximum (FWHM), respectively. Fragmentation in Q2 was conducted with 1.5 mTorr argon. Multiple-reaction monitoring (MRM) transitions corresponding to the neutral loss of a 2-deoxyribose (116 Da) from the $[M + H]^+$ of 5-mdC ($m/z$ 242 → 126), $[^{13}C_5]$-5-mdC ($m/z$ 247 → 126), dG ($m/z$ 268 → 152) and $[^{15}N_5]$-dG ($m/z$ 273 → 157) were monitored (Supplementary Data 1). Quantification of 5-mdC and dG (in moles) in the nucleoside mixtures were performed using peak area ratios for the analytes over their corresponding stable isotope-labeled standards, along with the fixed amounts of the added internal standards (in moles), and the calibration curves. The levels of 5mdC, reported as percentages of 5mdC relative to dG, were calculated by dividing the molar quantifies of 5-mdC with those of dG. The p-values were calculated using two-tailed, unpaired Student's t-test. (EV: $n = 6$; WT: $n = 12$; R181A: $n = 6$; F190A: $n = 6$). The source data are provided as a Source Data file.

### Statistics and reproducibility
The two-tailed Student t-test was performed to compare distributions between different groups. And the p-value <0.05 was considered to be statistically significant. The chromatin fractionation experiment was performed twice. No statistical method was used to predetermine sample size. No data were excluded from the analyses.

### Reporting summary
Further information on research design is available in the Nature Portfolio Reporting Summary linked to this article.

## Data availability
The consensus, local and composite 3D cryo-EM maps for the DNMT3A1 UDR–H2AK119ub1 NCP complex have been deposited in the Electron Microscopy Data Bank under the accession numbers EMD-41920, EMD-41921 and EMD-41922, respectively. Atomic coordinates for the structural models have been deposited in the Protein Data Bank under accession code 8U5H. The PDB accession codes 6PA7 and 6WKR were used in this study. Source data are provided as a Source Data file or Supplementary Fig. 10. Source data are provided with this paper.

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

## Acknowledgements

This work was supported by NIH grants (R35 GM119721 to J.S., R01 CA215284 and R01 CA211336 to G.G.W., and R35 ES031707 to Y.W.) and a fund of Duke School of Medicine to G.G.W. G.G.W. is a Leukemia & Lymphoma Society (LLS) Scholar. Cryo-EM data were collected at the National Center for CryoEM Access and Training (NCCAT) and the Simons Electron Microscopy Center located at the New York Structural Biology Center, supported by the NIH Common Fund Transformative High Resolution Cryo-Electron Microscopy program (U24 GM129539,) and by grants from the Simons Foundation (SF349247) and NY State Assembly. Molecular graphics and analyses were performed with UCSF ChimeraX, developed by the Resource for Biocomputing, Visualization, and Informatics at the University of California, San Francisco, with support from National Institutes of Health R01-GM129325 and the Office of Cyber Infrastructure and Computational Biology, National Institute of Allergy and Infectious Diseases.

## Author contributions

X.C., Y.G., and T.Z. performed the experiments, J.L. and J.F. assisted in sample preparation and protein binding assays. X.C. and J.S. performed structural analysis. Y.W., G.G.W. and J.S. supervised the study. X.C. G.G.W. and J.S. wrote the manuscript and all authors approved the manuscript.

## Competing interests

The authors declare no competing interest.
