## [Peer Review File · Nature Communications]

Structural basis for the H2AK119ub1-specific DNMT3A-nucleosome interactionReviewers' Comments:

Reviewer #1:

Remarks to the Author:

DNA methylation is a key epigenetic mark, and it is a very important question to understand how it is linked to other chromatin factors, in particular to the Polycomb complexes.

Two recent papers by the Allis and Goodell labs have made exciting inroads into this question (Weinberg Nat Genet 2021 PMID: 33986537, Gu Nat Genet 2022 PMID: 35534561), by showing that the N-terminal part of the de novo methyltransferase DNMT3A1 directs its colocalization with PRC1. Molecularely, an unstructured domain (UDR) is necessary and sufficient to bind the ubiquitinated histone H2AK119Ub, which is the product of PRC1 activity.

The current manuscript by Jikui Song and coworkers examines the structural basis of this phenomenon. The authors:

- perform BLI binding assays between the Nterminal domain of DNMT3A1 and nucleosomes that contain or lack Ub on H2AK119. The Kd is 200 nM without the Ub, and 16 nM with the Ub (Fig. 1).
- perform Cryo-EM of H2AK119Ub-containing NCPs bound to the UDR. They arrive at a 3 Å resolution, and visualize two molecules of UDR bound to each NCP (one on each H2AK119Ub, Fig.1).
- identify the UDR residues that mediate the interaction with the H2A/H2B acidic patch, and validate their importance by binding assays (Fig. 2)
- identify the UDR residues that mediate the interaction with the H2A Cterminal segment/H3 homodimeric interface, and validate their importance by binding assays (Fig. 3)
- identify the UDR residues that mediate the interaction with nucleosomal DNA and H2AK119Ub, and validate their importance by binding assays (Fig. 4)
- show that the UDR can outcompete the polycomb-associated protein JARID2 for binding to the NCP containing H2AK119Ub (Fig. 5)

The paper is well written. The experiments are well carried out and interpreted, and I see no obvious concern with them. However, I feel that this paper may be too underdeveloped for Nat Comm, and would be a better fit for a more specialized journal. The two obvious limitations of this work are:

- the authors perform structural work (and they do it well), but do not provide the biological significance of their findings. They could do it by generating mutant ES lines, as they have in other Nat Comm papers (Gao 2022 PMID: 35869095, Gao 2020 PMID: 32620778).
- the structure is limited to a small region of the UDR, as opposed to larger DNMT3A1 construct, which would teach us more about the intramolecular dynamics

Reviewer #2:

Remarks to the Author:

The authors investigated the interaction of the DNMT3A1 UDR domain with a nucleosome containing the H2A K119ub1 modification. They discovered a novel binding mode in which the phylogenetically conserved part of this domains deeply delves into the structure forming multivalent interactions with the nucleosome, DNA and ubiquitin. Mutational study validated these findings. Moreover, a competitive nucleosome binding of UDR and the N-terminal UIM of Jarid2 was identified, thereby providing a mechanistic explanation for the widely observed antagonism of DNA methylation and H3K27me3, which exists despite the H2AK119ub1 interaction of DNMT3A1. The paper reports highly important novel data the provide deep mechanistic information about critical and highly important epigenetic regulatory processes. All experiments were conducted at very high level and the manuscript is written very clear and thorough. I have only few comments for this excellent paper:

- 1) It would be important to describe the relationship of the newly described UDR structure and the previously published DNMT3A2/3B3 nucleosome structure. Is UDR binding feasible in the context of the DNMT3A2/3B3 complex? If not, can the authors propose a model of how these complexes are

related? In this context Fig. S7 could be further enhanced and edited for clarity.

2) I think it would be important for readers to mention the connection that the competitive binding of UDR and Jarid2 can explain the antagonism of DNA methylation and H3K27me3 briefly in the abstract and also in the short result summary at the end of the introduction.

3) The mutational data could be summarized in a table, specifying the structural role of the mutated residues, the K_d values, and the fold effect. A summary bar diagram of the fold-effects may also be interesting.

4) For the BLI data, handling of the repetitive measurements should be made more transparent. I presume, the line shows one exemplary experiment and the K_d values are average \pm SD. Please clarify.

5) In Fig. S5 the information must be added that UDR is shown in blue.

6) The authors may check if the term H2AK119ub1 is more appropriate.

Reviewer #3:

Remarks to the Author:

DNMT3A is a de novo DNA methyltransferase that involves in embryogenesis, gametogenesis and carcinogenesis. Human DNMT3A consists of two isoforms, DNMT3A1 and A2. These isoforms have been shown different expression patterns. DNMT3A1 is widely expressed in somatic cells and enriched in transcriptionally inactive heterochromatin. DNMT3A consists of N-terminal PWWP (H3K36me2/me3 binding), ADD (H3K4me0 binding) and methyltransferase domains. In addition, DNMT3A1 has a unique motif for recognizing monoubiquitinated histone H2AK119 (H2AK119ub), ubiquitin-dependent recruitment (UDR). However, molecular mechanism by which the UDR specifically recognizes H2AK119ub is unclear.

In this paper, the authors determined the cryo-EM structure of the DNMT3A1 UDR in complex with H2AK119ub-modified nucleosome core particle (NCP). The structural study unveiled the recognition of H2AK119ub by the UDR; The UDR interacts with acidic patch of the NCP, with groove formed by C-terminal of H2A and H3 dimer interface, and with nucleosomal DNA and ubiquitin moiety. This structural study successfully showed the molecular mechanism underlying H2AK119ub by the UDR. However, the reviewer has some concerns about the interpretation of the cryo-EM data and biochemical assay.

[Comments]

Cryo-EM map corresponding to the ubiquitin moiety is fairly low resolution judging from Supplementary Figure 2, 3c and 3d. Without any explanations in the manuscript, the authors determined the spatial arrangement of ubiquitin to fit to the cryo-EM map. The structural model indicated that I36 patch of ubiquitin interacted with the UDR residues. I wondered if the spatial orientation of ubiquitin was indeed correct. The authors should perform binding assay using H2AK119ub-modified NCP including I36 patch residues mutant of ubiquitin to demonstrate that the spatial orientation of ubiquitin determined by the authors is correct.

In addition to the binding assay, UDR-dependent DNA methylation of H2AK119ub-modified NCP is required for showing the importance of the interaction between UDR and H2AK119ub-modified NCP. DNA methylation assay using the structure-guided mutants of DNMT3A1 should be conducted.

General Response

We thank all three reviewers for their collective efforts in reviewing our manuscript and their positive view of our work. The reviewers pointed out that our work “well carried out and interpreted” (Reviewer 1), “conducted at very high level and the manuscript is written very clear and thorough” (Reviewer 2), and “successfully showed the molecular mechanism underlying H2AK119ub by the UDR” (Reviewer 3). In addition, the reviewers have raised a number of constructive comments to improve our manuscript. As outlined below in the point-by-point response (marked in blue), we have now systematically addressed all the raised critiques and have incorporated them in the revised manuscript (marked in red).

Point-by-point Response

Response to Reviewer 1

Reviewer #1 (Remarks to the Author):

DNA methylation is a key epigenetic mark, and it is a very important question to understand how it is linked to other chromatin factors, in particular to the Polycomb complexes. Two recent papers by the Allis and Goodell labs have made exciting inroads into this question (Weinberg Nat Genet 2021 PMID: 33986537, Gu Nat Genet 2022 PMID: 35534561), by showing that the N-terminal part of the de novo methyltransferase DNMT3A1 directs its colocalization with PRC1. Molecularly, an unstructured domain (UDR) is necessary and sufficient to bind the ubiquitinated histone H2AK119Ub, which is the product of PRC1 activity.

The current manuscript by Jikui Song and coworkers examines the structural basis of this phenomenon. The authors:

- perform BLI binding assays between the Nterminal domain of DNMT3A1 and nucleosomes that contain or lack Ub on H2AK119. The Kd is 200 nM without the Ub, and 16 nM with the Ub (Fig. 1).
- perform Cryo-EM of H2AK119Ub-containing NCPs bound to the UDR. They arrive at a 3 Å resolution, and visualize two molecules of UDR bound to each NCP (one on each H2AK119Ub, Fig.1).
- identify the UDR residues that mediate the interaction with the H2A/H2B acidic patch, and validate their importance by binding assays (Fig. 2)
- identify the UDR residues that mediate the interaction with the H2A Cterminal segment/H3 homodimeric interface, and validate their importance by binding assays (Fig. 3)
- identify the UDR residues that mediate the interaction with nucleosomal DNA and H2AK119Ub, and validate their importance by binding assays (Fig. 4)
- show that the UDR can outcompete the polycomb-associated protein JARID2 for binding to the NCP containing H2AK119Ub (Fig. 5)

The paper is well written. The experiments are well carried out and interpreted, and I see no obvious concern with them. However, I feel that this paper may be too underdeveloped for Nat Comm, and would be a better fit for a more specialized journal.

We thank Reviewer 1 for an accurate summary of our work and considering our paper “well written” and the experiments “well carried out and interpreted”. Reviewer 1 also raised a concern on the thoroughness of this work. In response, we have revised the manuscript with additional data and clarifications, as summarized below.

The two obvious limitations of this work are:

-the authors perform structural work (and they do it well), but do not provide the biological significance of their findings. They could do it by generating mutant ES lines, as they have in other Nat Comm papers (Gao 2022 PMID: 35869095, Gao 2020 PMID: 32620778).

We thank the reviewer for the comment and suggestion. As the reviewer pointed out, the recent ground-breaking findings by the Allis and Goodell labs elucidated the importance of the DNMT3A UDR-H2AK119ub1 interaction and its crosstalk with PRC1 signaling. The current work complements these studies by providing mechanistic insights into H2AK119ub1-specific DNMT3A1 recruitment and functional antagonism between DNA methylation and H3K27me3, as well as an explanation to the previously observed mutational effects on the UDR region. These advances shall appeal to the broad audience of chromatin biology and epigenetics.

We agree with the reviewer that generating mutant ES cell lines would further strengthen the functional significance of the observations. Indeed, this structural work provides a basis for further understanding of the interplay between DNMT3A domains, the functional divergence of DNMT3A1 and DNMT3A2, and the crosstalk between DNA methylation and polycomb signaling, all of which warrant future investigation. In the revised manuscript, we included additional discussion on the functional implication of this study and its link to the recent studies by Allis group and Goodwell group (line 387-396). We also consider that it is important to report this highly competitive and broad-impact work in a timely manner. Therefore, we respectfully request to consider the cellular analysis for a follow-up study.

-the structure is limited to a small region of the UDR, as opposed to larger DNMT3A1 construct, which would teach us more about the intramolecular dynamics

The reviewer’s point is well taken. Our consideration of focusing UDR, rather than full-length DNMT3A1, for structural study is based on the considerations that (i) recent studies by Allis and Goodwell labs indicated that there is a competing chromatin-binding mechanism between the DNMT3A1 UDR and PWWP domains, raising a challenge in structural characterization of a longer DNMT3A fragments and nucleosome at the mono-nucleosome level; and (ii) use of the well-defined UDR fragment ensures good sample quality and clear-cut data interpretation.

To alleviate the reviewer’s concern, and as suggested by Reviewer 3, we have introduced H2AK119ub1 interaction-deficient mutations to full-length DNMT3A1, including R181A, F190A and K202A, and performed *in vitro* DNA methylation kinetic assays. In comparison with wild-type DNMT3A1, R181A-, F190A- and K202A-mutated DNMT3A1 shows much reduced methylation

efficiency on nucleosome with a linker DNA harboring multiple CpG sites (Fig. R1). This observation, consistent with our structural study, suggests that the DNMT3A1 UDR-H2AK119ub1 interaction plays an important role in DNMT3A1-mediated DNA methylation under the nucleosome environment.

Figure R1. *In vitro* DNA methylation kinetics of full-length DNMTA1, WT or H2AK119ub1 binding-defective mutants. The nucleosome with linker DNA containing multiple CpG sites was used as substrates. This data has been included as Fig. 4p in the revised manuscript.

Response to Reviewer 2

Reviewer #2 (Remarks to the Author):

The authors investigated the interaction of the DNMT3A1 UDR domain with a nucleosome containing the H2A K119ub1 modification. They discovered a novel binding mode in which the phylogenetically conserved part of this domains deeply delves into the structure forming multivalent interactions with the nucleosome, DNA and ubiquitin. Mutational study validated these findings. Moreover, a competitive nucleosome binding of UDR and the N-terminal UIM of Jarid2 was identified, thereby providing a mechanistic explanation for the widely observed antagonism of DNA methylation and H3K27me3, which exists despite the H2AK119ub1 interaction of DNMT3A1. The paper reports highly important novel data the provide deep mechanistic information about critical and highly important epigenetic regulatory processes. All experiments were conducted at very high level and the manuscript is written very clear and thorough. I have only few comments for this excellent paper:

We are grateful to Reviewer 2 for his/her positive view of this study. In the revised manuscript, we have addressed the reviewer's concerns as below.

1) It would be important to describe the relationship of the newly described UDR structure and the previously published DNMT3A2/3B3 nucleosome structure. Is UDR binding feasible in the context of the DNMT3A2/3B3 complex? If not, can the authors propose a model of how these complexes are related? In this context Fig. S7 could be further enhanced and edited for clarity.

The reviewer raised an important point. Structural comparison of the UDR-nucleosome structure with that of DNMT3A2/3B3-nucleosome indicates that the UDR-nucleosome interaction would interfere with the DNMT3B3-nucleosome contact, both of which involve the acidic patch of the nucleosome. Following the reviewer's comment, we have included a structural superposition of

the nucleosome-contact between DNMT3A1 UDR and DNMT3A2/B3 in Fig. S9 in the revised manuscript.

2) I think it would be important for readers to mention the connection that the competitive binding of UDR and Jarid2 can explain the antagonism of DNA methylation and H3K27me3 briefly in the abstract and also in the short result summary at the end of the introduction.

We thank the reviewer for the suggestion and have included this point in the abstract and the introduction in the revised manuscript.

3) The mutational data could be summarized in a table, specifying the structural role of the mutated residues, the Kd values, and the fold effect. A summary bar diagram of the fold-effects may also be interesting.

We thank the reviewer for this excellent suggestion. We have included a table as well as a bar diagram summarizing the fold-effects of the mutations on the binding in the revised manuscript.

4) For the BLI data, handling of the repetitive measurements should be made more transparent. I presume, the line shows one exemplary experiment and the Kd values are average +- SD. Please clarify.

In the revised manuscript, we have clarified that the line graph represents one experiment, and the Kd values were derived from two independent measurements.

5) In Fig. S5 the information must be added that UDR is shown in blue.

We have included such information in Fig. S4 in the revised manuscript.

6) The authors may check if the term H2AK119ub1 is more appropriate.

We agree with the reviewer and have replaced H2AK119ub with H2AK119ub1 for accuracy in the revised manuscript.

Response to Reviewer 3

Reviewer #3 (Remarks to the Author):

DNMT3A is a de novo DNA methyltransferase that involves in embryogenesis, gametogenesis and carcinogenesis. Human DNMT3A consists of two isoforms, DNMT3A1 and A2. These isoforms have been shown different expression patterns. DNMT3A1 is widely expressed in somatic cells and enriched in transcriptionally inactive heterochromatin. DNMT3A consists of N-terminal PWWP (H3K36me2/me3 binding), ADD (H3K4me0 binding) and methyltransferase domains. In addition, DNMT3A1 has a unique motif for recognizing monoubiquitinated histone H2AK119 (H2AK119ub), ubiquitin-dependent recruitment (UDR). However, molecular mechanism by which the UDR specifically recognizes H2AK119ub is unclear. In this paper, the authors determined the cryo-EM structure of the DNMT3A1 UDR in complex with H2AK119ub-modified nucleosome core particle (NCP). The structural study unveiled the

recognition of H2AK119ub by the UDR; The UDR interacts with acidic patch of the NCP, with groove formed by C-terminal of H2A and H3 dimer interface, and with nucleosomal DNA and ubiquitin moiety. This structural study successfully showed the molecular mechanism underlying H2AK119ub by the UDR. However, the reviewer has some concerns about the interpretation of the cryo-EM data and biochemical assay.

We thank the reviewer for the positive view of this work. We also thank the reviewer for raising the comments that improved this manuscript.

[Comments]

Cryo-EM map corresponding to the ubiquitin moiety is fairly low resolution judging from Supplementary Figure 2, 3c and 3d. Without any explanations in the manuscript, the authors determined the spatial arrangement of ubiquitin to fit to the cryo-EM map. The structural model indicated that I36 patch of ubiquitin interacted with the UDR residues. I wondered if the spatial orientation of ubiquitin was indeed correct. The authors should perform binding assay using H2AK119ub-modified NCP including I36 patch residues mutant of ubiquitin to demonstrate that the spatial orientation of ubiquitin determined by the authors is correct.

The reviewer raised an important point. Following the reviewer's suggestion, we have introduced mutations to residues I36 and L71 of ubiquitin and performed BLI binding assays. As shown in Fig. R2, introducing I36A or L71A mutation to H2AK119ub1 each decreased the binding of DNMT3A1 UDR–H2K119ub1 nucleosome by ~1.5 fold, supporting the structural observation that the UDR residues contact the I36 patch of H2AK119ub1.

Figure R2. BLI binding assays for DNMT3A1 UDR with H2AK119ub1-mutated NCP. This data has been included as Fig. 4n,o in the revised manuscript.

In addition to the binding assay, UDR-dependent DNA methylation of H2AK119ub-modified NCP is required for showing the importance of the interaction between UDR and H2AK119ub-modified NCP. DNA methylation assay using the structure-guided mutants of DNMT3A1 should be conducted.

Following the reviewer's comments, we have introduced several H2AK119ub1 binding-defective mutations, including R181A, F190A and K202A, into full-length DNMT3A1 and performed *in vitro* DNA methylation kinetic assays. Our results indicated that introducing the H2AK119ub1 binding-defective mutations each markedly impairs DNMT3A1-mediated DNA methylation, with the methylation efficiency reduced by 1.6-2.6 folds after a 30-min reaction (Fig. R1), suggesting that

the interaction with H2AK119ub1-marked nucleosome critically underpins DNMT3A1-mediated DNA methylation in the nucleosome environment. We thank the reviewer for this suggestion.

Reviewers' Comments:

Reviewer #1:

Remarks to the Author:

I thank the authors for their revision.

I feel disappointed that my main request, ie the generation of mutant ES lines, has not been addressed experimentally. My personal impression was that this kind of in vivo validation was necessary to reach the level of a Nat Comm. However, Dr Hua, the editor, is obviously the better judge of this.

Other than that, as I said before, the work is well carried out, and the additional panel generated by the author is a useful addition.

Reviewer #2:

Remarks to the Author:

This is a very nice paper. My comments have been convincingly addressed and (as far as I can judge) this is also true for the other reviewers' comments. I recommend acceptance of this work. It is a strong contribution with important finding and potentially strong impact in the field.

Reviewer #3:

Remarks to the Author:

The authors have appropriately revised the manuscript according to the reviewer's suggestion. There are no further comments with the manuscript.

We thank all three reviewers for their positive assessment of our manuscript. As outlined below in the point-by-point response (marked in blue), we have now addressed the remaining critique of Reviewer #1 and have incorporated them in the revised manuscript (marked in red).

Reviewer #1 (Remarks to the Author):

I thank the authors for their revision.

I feel disappointed that my main request, ie the generation of mutant ES lines, has not been addressed experimentally. My personal impression was that this kind of in vivo validation was necessary to reach the level of a Nat Comm. However, Dr Hua, the editor, is obviously the better judge of this.

We appreciate the Reviewer's view on the importance of cellular validation of our work. Following the reviewer's suggestion, we transduced exogenous DNMT3A1, either WT or UDR mutant (R181A or F190A), to mouse ESCs with triple knockout (TKO) of Dnmt1, Dnmt3a and Dnmt3b.

As shown in Fig. R1, our immunofluorescence assay reveals that the F190A mutation of DNMT3A1 led to a diffuse pattern, reminiscent of what was previously observed for DNMT3A2, therefore supporting a link between the UDR-nucleosome interaction and cellular localization of DNMT3A1. On the other hand, the other UDR mutation, R181A, did not appreciably alter the cellular localization of DNMT3A1, suggesting that the R181 site alone is not sufficient to confer the DNMT3A1-unique cellular localization. Consistent with the IF analysis, our chromatin fraction assay indicated that the F190A mutation, but not the R181A mutation, substantially reduces the chromatin association of DNMT3A1, leading to markedly reduced genome methylation (Fig. R1d,e).

While we've conducted the experiment requested by Reviewer 1 (e.g. the rescue using the TKO mESC), our in vitro and in vivo results also point to rather complex regulation of DNMT3A1. We believe that our extensive structural and biochemical studies of DNMT3A1 are solid and timely, provide the critical initial glimpse into the DNMT3A1 UDR-directed functional regulation and thus warrants publication.

To ease the reviewer's concern, we've included more discussion at the related Results sections, as well as a paragraph of "limitations of this study" in the end of the Discussion section. Here, we surmise that some unknown mechanism and/or unknown cellular factor might play a role in coordinating with the UDR-nucleosome interaction to regulate the cellular activity of DNMT3A1. In support of this notion, previous studies have indicated that the heterochromatin binding of DNMT3A1 is also regulated by other chromatin factors, such as SETDB1 and HP1 (refs. 42-44). In addition, the AlphaFold-based structural

modeling of full-length DNMT3A1 reveals potential intramolecular interactions between the UDR and ADD domains of DNMT3A1, with the UDR occupying the entire H3-binding site of the ADD domain (Fig. R2). Whether and how the UDR-nucleosome interaction interplays with other intra- or inter-molecular interactions for regulating the functionality of DNMT3A1 is beyond the scope of this current work and warrants a future study. We thank the reviewer for the suggestion.

Figure R1. Effect of the UDR-nucleosome interaction on the activity of DNMT3A in cells. (a) Western blotting shows the expression level of DNMT3A variants in TKO cells. The molecular weight markers are labeled on the right. (b) Immunofluorescence analysis for cellular localization of DNMT3A1, WT or mutant, and its overlap with DAPI foci. Scale bar, 5 μ m. (c) box plot shows correlation coefficient between DNMT3A and DAPI. The whiskers are minimum to maximum, the box depicts the 25th–75th percentiles, and the line in the middle of the box is plotted at the median. (n = 20) Two-sided unpaired t-test was

used for statistical analysis. P-values are labeled on top of each comparison. (d) Chromatin fractionation assay shows the localization and chromatin binding affinity of different DNMT3A1 variants. F190A, but not the WT or R181A mutation, led to the DNMT3A distribution into cytoplasm and weakened its binding to chromatin in nucleus. (e) Mass spectrometry-based quantification for global DNA methylation levels in TKO cells with stable expression of the indicated DNMT3A. P-values are labeled on top of each comparison. (WT: n=12; EV: n=6; R181A: n=6; F190A: n=6). These data have been included as Fig. 5 in the revised manuscript.

Figure R2. Structural modeling analysis of a potential intramolecular interaction of DNMT3A1. (a) Ribbon representation of the DNMT3A ADD domain (slate) bound the H3 peptide (yellow). (PDB 3A1B). (b) AlphaFold model of DNMT3A1 indicating a potential intramolecular interaction between the UDR (cyan) and the ADD domain (lightpink). Zinc atoms in (a,b) are shown in sphere representation.

Other than that, as I said before, the work is well carried out, and the additional panel generated by the author is a useful addition.

We thank the reviewer for the positive assessment of our revision.

Reviewer #2 (Remarks to the Author):

This is a very nice paper. My comments have been convincingly addressed and (as far as I can judge) this is also true for the other reviewers' comments. I recommend acceptance of this work. It is a strong contribution with important finding and potentially strong impact in the field.

We thank the reviewer for his/her support of this work.

Reviewer #3 (Remarks to the Author):

The authors have appropriately revised the manuscript according to the reviewer's suggestion. There are no further comments with the manuscript.

We thank the reviewer for his/her support of this work.

Reviewers' Comments:

Reviewer #1:

Remarks to the Author:

I thank the authors for their efforts with this revision. I now support acceptance of the revised manuscript.

Reviewer #1 (Remarks to the Author):

I thank the authors for their efforts with this revision. I now support acceptance of the revised manuscript.

We thank the reviewer for his/her support of publishing our manuscript.